# Evaluation and uncertainty analysis of regional scale CLM4.5 net carbon flux estimates

Hanna Post[1,2,3], Harrie-Jan Hendricks Franssen[1,3], Xujun Han[1,3], Roland Baatz[1,3], Carsten Montzka[1], Marius Schmidt[1], Harry Vereecken[1,3]

1) Agrosphere (IBG-3), Forschungszentrum Jülich GmbH, 52425 Jülich, Germany;

2) Institute of Geography, University of Cologne, Cologne, Germany

3) Centre for High-Performance Scientific Computing in Terrestrial Systems: HPSC TerrSys, Geoverbund ABC/J, Leo-Brandt-Strasse, 52425 Jülich, Germany

*Correspondence to*: Hanna Post (h.post@uni-koeln.de)

**Abstract.** Modelling net ecosystem exchange (NEE) at the regional scale with land surface models (LSMs) is relevant for the estimation of regional carbon balances, but studies on that are very limited. Furthermore, it is essential to better understand and quantify the uncertainty of LSMs in order to improve LSMs. An important key variable in this respect is the prognostic leaf area index (LAI), which is very sensitive to forcing data and strongly affects the modelled NEE. We applied the Community Land Model (CLM4.5-BGC) to the Rur catchment in western Germany and compared estimated and default ecological key parameters for modelling carbon fluxes and LAI. The parameter estimates were previously estimated with the Markov Chain Monte Carlo (MCMC) approach DREAM$_{(zs)}$ for four of the most widespread plant functional types in the catchment. It was found that the catchment scale annual NEE was strongly positive with default parameter values but negative (and closer to observations) with the estimated values. Thus, the estimation of CLM parameters with local NEE observations can be of high relevance when determining regional carbon balances. To obtain a more comprehensive picture of model uncertainty, CLM ensembles were set up with perturbed meteorological input and uncertain initial states in addition to uncertain parameters. C3-grass and C3-crop were particularly sensitive to the perturbed meteorological input, which resulted in a strong increase of the standard deviation of the annual NEE sum ($\sigma_{\sum NEE}$) for the different ensemble members from ~2-3 gC m$^{-2}$ y$^{-1}$ (with uncertain parameters) to ~45 gC m$^{-2}$ y$^{-1}$ (C3-grass ) and ~75 gC m$^{-2}$ y$^{-1}$ (C3-crop) with perturbed forcings. This increase of uncertainty is related to the impact of the meteorological forcings on leaf onset and senescence, and enhanced/reduced drought stress related to perturbation of precipitation. The NEE uncertainty for the forest PFT was considerably lower ($\sigma_{\sum NEE}$ ~ 4.0 -13.5 gC m$^{-2}$ y$^{-1}$ with perturbed parameters, meteorological forcings and initial states).We conclude that LAI and NEE uncertainty with CLM is clearly underestimated if uncertain meteorological forcings and initial states are not taken into account.

# 1. Introduction

Net ecosystem $CO_2$ exchange (NEE), the difference of $CO_2$ release via soil and plant respiration and photosynthetic $CO_2$ uptake, is an important indicator for the net carbon sink or source function of terrestrial ecosystems. The understanding of factors controlling the spatial and temporal variability of carbon fluxes like respiration is still limited (Reichstein and Beer, 2008). Eddy covariance (EC) net carbon flux measurements are limited to a relatively small area. Chen et al. (2012) showed that the 90% cumulative annual footprint area of 12 EC towers located at Canadian sites (with different land cover including grassland and forest) varied from about 1.1 $km^2$ to 5.0 $km^2$, and that the spatial representativeness of the EC flux measurements depends on the degree of the land surface heterogeneity. Biogeochemical fluxes are spatially and temporally highly variable and nonlinear due to the spatial heterogeneity of soil properties, vegetation, and fauna, and the temporal variability of the environmental drivers (e.g. meteorological conditions, management schemes)  (Chen et al., 2009; Stoy et al., 2009; Borchard et al. 2015). Therefore, conventional interpolation methods like for example kriging or inverse distance weighting are not suited to upscale EC carbon flux measurements to larger areas.

Land surface models like CLM (Oleson et al., 2013) simulate the coupled carbon, nitrogen, water and energy cycle of the land surface, and are essential to understand interactions between the climate and the terrestrial carbon cycle and to predict climate-ecosystem feedbacks (e.g. Quéré et al., 2012; Arora et al., 2013; Brovkin et al., 2013; Todd-Brown et al., 2014). In this study, CLM version 4.5 in the biogeochemistry (BGC) mode (CLM4.5-BGC) was applied. The prognostically calculated leaf area index (LAI) is a major indicator for the model representation of plant phenology. Moreover, it affects photosynthesis and transpiration. Therefore, the LAI is a key state variable for carbon flux predictions and land surface-atmosphere exchange fluxes of water and carbon. Thus, a correct representation of the simulated LAI in terms of magnitude and timing is highly desirable. The LSM representation of plant phenology (timing of plant emergence, length in growing season) was shown to be seriously flawed in LSMs (e.g. Richardson et al. 2012) including CLM (Dahlin et al. 2015), which can also affect carbon flux estimates (Baldocchi and Wilson, 2001; Richardson et al., 2012). Simulated carbon fluxes and the prognostic LAI in CLM4.5-BGC are closely linked, because they depend on common ecological key parameters and plant phenology schemes.

Commonly, LSMs are applied at global or continental scale (e.g. Stöckli et al., 2008; Bonan et al., 2011; Lawrence et al., 2012) with grid sizes of ~ 0.25°- 1.5°. At this coarse scale with such a high degree of spatial aggregation, and given the non-linearity of the governing equations, the modelled values for output variables like NEE can deviate strongly from the ones that would have been calculated with a fine resolution model using fine resolution input. Moreover, a reliable calibration and validation of global LSMs is difficult, because observed data including soil carbon stocks and EC fluxes are only available for single locations. When applying a LSM for a small region or catchment with a high spatial resolution (e.g. 1 $km^2$, as in this study) the error of simulated fluxes is expected to be smaller due to the lower degree of spatial aggregation (Anderson et al., 2003). Besides, the land cover within a 1 $km^2$ grid cell more likely matches with the land cover at the EC site, which

enables a grid-based evaluation of modelled NEE. A high spatial resolution can better represent the land surface heterogeneity and regional weather variability than a coarse spatial resolution. Thus, regional or catchment scale applications of LSMs allow for investigating spatial patterns of model states, biogeochemical fluxes and interactions with the regional climate and catchment hydrology. Accordingly, quantification of carbon fluxes at regional scales can enhance the understanding of $CO_2$ dynamics and their drivers (Desai et al., 2008). This has been shown in various studies e.g. for West Africa (Bonan et al., 2002; Li et al., 2007) and the Alaskan Arctic (Fisher et al., 2014). However, to our knowledge, studies like Xiao et al. (2011) who optimized a simple ecosystem model to upscale measured EC carbon fluxes to the regional scale do not exist yet for more complex LSMs like CLM. This is because (i) high resolution input data is often not available, (ii) the implementation of a new model set-up to a specific region is relatively time consuming, and (iii) careful parameter estimation is required to allow for meaningful predictions.

LSM predictions of carbon, water and energy fluxes are still subject to a high degree of uncertainty due to (i) model structural deficits related to an imperfect and incomplete model representation of the biogeochemical processes (Todd-Brown et al., 2012; Foereid et al., 2014), (ii) poorly constrained model parameters (Abramowitz et al., 2008; Beven and Freer, 2001; Todd-Brown et al., 2013), (iii) errors in the representation of initial model conditions generated via a spin-up (Carvalhais et al., 2010; Kuppel et al., 2012; Xia et al., 2012; Exbrayat et al., 2014a), as well as (iv) errors in both atmospheric and land surface input data. Some studies estimate the uncertainty of terrestrial carbon flux predictions based on an ensemble of many different LSMs (e.g. Fisher et al., 2014; Huntzinger et al., 2012; Piao et al., 2013; Zhao et al., 2016). These studies focus on differences between models and therefore model structural deficits. Those studies highlight that (i) carbon flux predictions are generally highly uncertain, which also contributes to the uncertainty in climate change predictions, (ii) interactions of the different processes and drivers is not understood satisfactorily, and (iii) models require structural improvement to produce more consistent predictions. In order to improve LSM model structure and thus model-data and inter-model consistency, a more comprehensive understanding of model functionality and the contribution and link of the different model error sources is required. However, as highlighted by Xiao et al. (2014), the uncertainty of carbon fluxes obtained by ecosystem and land surface models has not been analyzed and quantified enough, particularly in regional scale studies.

Whereas the uncertainty of land surface model parameters has been subject to intensive investigations (e.g., Ren et al., 2013; Xiao et al., 2014), and several works are dedicated to reducing this uncertainty, for example by parameter estimation with data assimilation methods (e.g., Safta et al., 2015), the other sources of uncertainty are less studied. It was already concluded in earlier work that parameter uncertainty alone cannot explain observed deviations between measured and simulated NEE (e.g., Pridhoko et al., 2008; Wang et al., 2011), and the remaining deviations are often attributed to model structural errors. However, also uncertainty in atmospheric forcings and initial conditions could contribute to unexplained deviations between simulated and measured NEE. From global scale studies it is known that NEE is sensitive to the climate scenario. For example, in a study the LSM LPJ-GUESS was forced with output from 18 different coupled atmosphere-ocean global

circulation models or earth system models, and showed very different NEE-responses depending on the climate scenario. Ten out of 18 models project that the land will become a carbon source in the 21$^{st}$ century, while the other eight models indicate that the land will act as a carbon sink (Ahlstrom et al., 2012). At the plot, catchment or regional scale and for shorter time periods in the recent past, very few studies analyzed the impact of uncertainty in meteorological forcings on NEE.

Studies which analyzed interannual variability in NEE detected the important role of temperature and precipitation as drivers of this variability (e.g., Keppel-Aleks et al., 2014). Gu et al. (2016) found that for the Missouri Ozark AmeriFlux forest site, the interannual NEE-variability is smaller in simulations by CLM than in the data. On the other hand, some field studies for pairs of monitoring sites (Kwon et al., 2006) or experimentation sites (with temperature increase and precipitation increase or decrease) (Xu et al., 2016) found a limited impact of temperature and precipitation differences (or changes) on NEE.

Zhang et al. (2012) is one of the few studies that analyzed in more detail the role of uncertain meteorological forcings (together with parameter uncertainty) for calculating NEE with a process-based ecosystem model. Their simulations for a Korean pine mixed forest site with two different input meteorological datasets revealed clear differences in model response. Spadavecchia et al. (2011) performed a rigorous uncertainty analysis for the combination of parameter and forcing uncertainty with a simple LSM for a pine stand in Oregon, USA. They found that the contribution of parameter uncertainty

to NEE-uncertainty is larger than the contribution of forcing uncertainty, but forcing uncertainty also has a significant impact. If meteorological stations are located more than 100km away from the study site, forcing uncertainty starts to dominate over parameter uncertainty. The relative contribution of initial state uncertainty to prediction uncertainty of LSMs has not been subject of intensive study although it was recognized as one of the major challenges in model-data fusion studies (Williams et al., 2009). Richardson et al. (2010) performed a Monte Carlo type inverse modelling to assimilate many

different data types in a forest carbon cycle model for the Howland AmeriFlux-site in Maine, USA. They estimated jointly model parameters and initial carbon pools. Although they found that model performance improved strongly, they also concluded that carbon pools are more difficult to constrain than ecosystem parameters. Peylin et al. (2016) also updated jointly parameters and carbon pools, in this case with the land surface model ORCHIDEE for a large scale application. These last two studies explicitly considered initial state uncertainty together with other uncertainty sources, but did not focus on the

relative contribution of the different uncertainty sources. As considering atmospheric forcings and initial states uncertainty, besides parameter uncertainty, has not received so much attention in the literature until now, we investigate the uncertainty of model predictions in relation to uncertain model parameters, initial conditions and atmospheric forcings in this work.

Accordingly, this study follows two main objectives. The first objective was to investigate to what extend Markov Chain Monte Carlo (MCMC) based parameter estimates can enhance the model-data consistency of NEE and LAI for a small

European catchment. The model used was CLM4.5 and compared with EC-measurements and LAI from the RapidEye satellite. We applied successfully validated parameter estimates from Post et al. (2016) for C3-grass, evergreen coniferous forest and broadleaf deciduous forest. For C3-crop, we estimated a new set of parameters as the estimated parameters by Post et al. (2016) did not improve the model-data fit in verification experiments, using the DiffeRential Evolution Adaptive

Metropolis DREAM (Vrugt, 2015). The main advantage of a multi-chain MCMC approach like DREAM is that it is not limited to to Gaussian distributed states and parameters and its robustness to find the global minimum. The model performance with the new parameter estimates was then compared to a reference run with global default parameters. The second objective was to investigate the uncertainty of modelled NEE and LAI resulting from uncertain model parameters, atmospheric forcings and initial states. We setup three different ensembles with different combinations of perturbed model input data in order to obtain insight into the contribution of these three main sources of model uncertainty to the uncertainty of the final model output.

## 2.    Data and Methods

### 2.1.    The Rur catchment

The Rur catchment (Figure 1) is located in the Belgian-Dutch-German border region and covers an area of 2354 km$^2$. It is characterized by two distinctly different areas of land use and climate. In the northern lowland part, precipitation amounts are lower (650-850 mm/a), and potential evapotranspiration is higher (580-600 mm/a) compared to the mountainous Eifel region in the south where annual precipitation is 850-1300 mm/a and potential evapotranspiration is 450-550 mm/a (Montzka et al., 2008a, 2008b). The annual mean temperature in the catchment ranges from about 7.5°C in the south of the catchment to about 10.3°C in the north (Baatz et al., 2014). The northern part is dominated by fertile loess soils and is intensely used for agriculture. Sugar beet and cereals (winter wheat, barley) are the most cultivated crops in the catchment (Figure 1). In the mountainous southern part shallow, less fertile soils predominate. It is mainly covered by meadows and forests. The Rur catchment is one of four central research regions of the TERENO project (Zacharias et al., 2011; Hohlfeld et al., 2012). TERENO has the main goal to determine global change impacts across different terrestrial compartments at the regional level. Therefore, comprehensive input and evaluation data is available for the catchment, including information on land use (Lussem and Waldhoff, 2013), leaf area indices (Ali et al., 2015; Reichenau et al., 2016) and EC data (Schmidt et al., 2012; Graf et al., 2014; Kessomkiat et al., 2013; Post et al., 2015).

### 2.1.1.    Eddy covariance data

Eddy covariance (EC) data in the Rur catchment were measured for four C3-crop sites for the evaluation periods summarized in Table 1. The EC crop sites Merzenhausen (ME), Niederzier (NZ), Selhausen (SE), and Engelskirchen (EN) are located in the northern lowland region of the catchment. In ME and SE winter wheat was grown during the measurement period, in EN and NZ sugar beet. The EC towers in ME and SE were permanently installed. For those sites EC data were available for more than one year. EC data at EN and NZ were measured by a roving station, which was installed two to three months at each of the three sites (Table 1). The complete processing of the raw data was performed with the TK3.1 software (Mauder and Foken, 2011),  using the quality flagging and uncertainty estimation scheme by Mauder et al., 2013 as outlined

in Post et al. (2015). Only non-gap filled data with quality flag 0 (high quality data) and 1 (moderate quality data) were included in this study.

Rollesbroich (RO) is and extensively used grassland site (Korres et al., 2010; Post et al (2015). Wüstebach (WÜ) (Graf et al. 2014) is located in the Eifel national park and is largely covered by evergreen coniferous forest, particularly spruce. RO and WÜ are the sites parameter were estimated for and validated in a previous study (Post et al. 2016), which were adopted here.

### 2.1.2. RapidEye-based leaf area index

RapidEye is a commercial satellite mission initiated by RapidEye AG (Tyc et al., 2005) and consists of five identical satellites, which were launched in August 2008. RapidEye provides multi-spectral images of five spectral bands (blue, green, red, red edge and near infra-red). The nominal temporal resolution is daily. The ground sampling distance is 6.5 m and the pixel size is 5 m for the orthorectified Level 3A data used here. The LAI data derived from satellite images are determined based on the NDVI (Normalized Difference Vegetation Index), which is related to the chlorophyll content in leaves. The NDVI is calculated based on the reflectances at near infra-red (NIR) and red (RED). NDVI-based LAI data ($LAI_{NDVI}$) are affected by various error sources, which can result in a high uncertainty of the LAI estimate. The major error sources are summarized in Garrigues et al. (2008), such as (i) uncertainties in surface reflectance measurements resulting e.g. from calibration errors or cloud contamination, or (ii) deficiencies in the representation of canopy architecture in the algorithms applied for the LAI retrieval, e.g. the negligence of foliage clumping. This can lead to a high underestimation of actual LAI, especially for needleleaf forests (Chen et al., 1997). Clumping effects on landscape scale are also related to the fact that LAI algorithms have been calibrated at the plot scale, but are applied over larger heterogeneous pixels, which can induce substantial scaling biases on the LAI estimates (Garrigues et al., 2006). The latter error source is assumed relatively small for the LAI retrieval from RapidEye due to the high spatial resolution (5m) of the images. Studies on verification or uncertainty quantification of LAI data derived from satellite images are very rare (Garrigues et al., 2008), but very important for land surface model applications. Ali et al. (2015) used orthorectified and radiometrically corrected Level 3A data (Blackbridge, 2015) to generate 5 m resolution LAI data for the Rur catchment, with the same methodology previously applied to MODIS data (e.g. Propastin and Erasmi, 2010). Those LAI data were validated for two crop sites (Merzenhausen and Selhausen) in the Rur catchment using in situ data measured with a destructive, ground-based method (Bréda, 2003) at several equally distributed points within the fields at six and eleven days during the growing season. The results indicate a high consistency between in situ measured LAI and $LAI_{NDVI}$ derived from RapidEye (Ali et al., 2015). Because only the two crop sites were included in this evaluation approach, the LAI data for crop sites (winter wheat) are considered most reliable. For this study, the $LAI_{NDVI}$ data for the Rur catchment obtained according to Ali et al. (2015) were aggregated from the 5 $m^2$ to the 1 $km^2$ grid of the CLM Rur catchment domain by arithmetically averaging.

## 2.2. Community Land Model set-up

The Community Land Model (CLM) version 4.5 (Oleson et al., 2013) with the active biogeochemistry model (CLM4.5-BGC) is fully prognostic with respect to the seasonal timing of vegetation growth and litter fall. The day length, the soil and air temperature as well as the soil water content are main determinants of plant phenology. Plant phenology representation follows three different schemes depending on the particular plant functional type (PFT): 1. evergreen phenology, 2. Seasonal deciduous phenology, and 3. Stress deciduous phenology. The four most widespread CLM-PFTs in the Rur catchment are: (1) needleleaf evergreen temperate trees, (2) broadleaf deciduous temperate trees, (3) C3 non-arctic grass, and (4) C3-crop. The average percentage PFT coverage of the vegetated land in Rur catchment was ~34% C3-crops, ~32% grassland, ~17% broadleaf deciduous forest and ~14% coniferous forest. Evergreen coniferous trees follow phenology scheme 1, deciduous broadleaf trees follow scheme 2, and both C3-grass and C3-crop follow scheme 3 (Oleson et al., 2013). The LAI and all carbon and nitrogen state variables in the vegetation, litter, and soil organic matter are calculated prognostically. NEE in CLM is calculated as the sum of gross primary production (GPP) and total ecosystem respiration (ER). Ecosystem respiration includes heterotrophic respiration (HR) and autotrophic respiration, the sum of maintenance respiration (MR) and growth respiration (GR) (Oleson et al., 2013). Photosynthesis is determined at leaf scale (Dai et al., 2004; Thornton and Zimmermann, 2007) and is upscaled by means of the LAI.

The definition of land use cover in CLM follows a nested sub-grid hierarchy structure (Oleson et al., 2013). The main land units, which are defined as percentage coverage per grid cell are: glacier, wetland, vegetated land, lake and urban area. Each land unit follows a different sub model scheme to calculate the respective carbon, water and energy fluxes for a certain grid cell. Each vegetated land unit has 15 soil columns and can include different plant functional types. The PFTs are also defined as percentage area of the vegetated area within the grid cell.

To apply CLM for the Rur catchment domain, a land surface input dataset was generated with a spatial resolution of 1km$^2$. The land unit for each grid cell and the PFT distribution of each vegetated land unit were defined based on the land use classification derived from supervised, multi-temporal remote sensing data analysis using RapidEye and ASTER data (Waldhoff et al., 2012; Lussem and Waldhoff, 2013). In addition to the land use coverage, CLM requires information on the percentages clay and sand for each of the 15 soil columns of the vegetated area per grid cell. For each soil layer, the soil texture was defined based on the German soil map (BK50) provided by the Geological Survey of North Rhine-Westphalia. Mean topographic slope, mean elevation and maximum fractional saturated area were determined for the 1 km$^2$ grid from a 10 m resolution digital elevation model (scilands GmbH, 2010). Additional land surface data required to run CLM4.5 such as soil color was adopted from the default CLM4.5 0.9° x 1.25° resolution global land surface data file of year 2000 (surfdata_0.9x1.25_simyr2000_c110921.nc).

The atmospheric forcing data used to run CLM are hourly time series of precipitation (mm/s), incoming short wave radiation (W m$^{-2}$), incoming long wave radiation (W m$^{-2}$), atmospheric pressure (Pa), air temperature (K), specific humidity (kg/kg)

and wind speed (mm/s) at the lowest atmospheric level. The data were obtained for the years 2008-2013 from the reanalysis COSMO-DE dataset provided by the German Weather Service (DWD) in 2.8 km x 2.8 km resolution (Baldauf et al., 2009). The COSMO-DE data were downscaled to 1 km$^2$ using natural neighbor interpolation based on Delaunay triangulation.

To generate the initial state variables such as the carbon and nitrogen pools, CLM was spun up over a period of 1200 years, using COSMO-DE data of the years 2008-2010. According to http://www.dwd.de/DE/klimaumwelt/klimaatlas/ klimaatlas_node.html, the annual average temperature in the years 2008 and 2009 was ~ 0.5-1.0 °C higher than the long term average (1961-1990) and the mean annual temperature in 2010 was ~ 0.5-1.0 °C lower. Mean precipitation amounts and freezing days were representative for the long term average. We also studied the effect of the forcing data used for the model spin-up. For example, we tested using a longer time series (1998-2004) of global climate data. However, we found that using more recent, regional forcing data during the spin-up resulted in carbon- and energy fluxes in better agreement with the observations. The model states obtained after the 1200-year spin-up were then used as input for a second three years "exit spin-up" also using the meteorological data for the years 2008-2010. The "exit spin-up" in CLM is necessary for technical reasons and switches the CLM settings from the (accelerated) spin-up mode to the "normal" mode in terms of the calculated carbon-nitrogen cycling. . A longer exit spin-up period of 100 years was tested, but results and model states differed only very minor from the case with a 3 years exit spin-up.

## 2.3. Parameter estimation and evaluation of model performance

### 2.3.1. Parameter estimation with DREAM$_{(zs)}$

In this study, we estimated the posterior probability density functions (pdfs) of five PFT-specific CLM45 parameters for C3-crop using the adaptive Markov Chain Monte Carlo (MCMC) method DREAM$_{(zs)}$ (Ter Braak and Vrugt, 2008; Laloy and Vrugt, 2012; Vrugt, 2015), according to the approach presented in Post et al. (2016). DREAM$_{(zs)}$ is based on the Bayes theorem (A1). In multi-chain MCMC methods like DREAM$_{(zs)}$, different (in our case three) Markov chains generate random walks through the parameter space and successively visit solutions with stable frequencies stemming from a stationary distribution. It is assumed herein that the prior distribution is uniform (non-informative) using ranges of the predefined upper and lower bounds for each parameter according to Post et al. (2016). The use of multiple chains offers a robust protection against premature convergence and thus the probability to become stuck in local minima is considerably reduced compared to single chain methods (Vrugt, 2015). The convergence of the parameters is monitored with the $\hat{R}$-convergence diagnostic of Gelman and Rubin (1992) which is computed for each dimension as the ratio of variance within one chain and the variance between different chains. Convergence is achieved if $\hat{R}$ is smaller than 1.2 for all parameters. Because every point in the parameter space is hit with a frequency proportional to its probability, the random walk allows to iteratively find a stable posterior distribution. Hence, after convergence, the density of the acceptance points in the parameter space approximates the posterior pdf according to Bayes theorem (see appendix). DREAM exhibits excellent sampling efficiencies

for multi-modal and high-dimensional posterior distributions (Vrugt, 2015), which is an important advantage if applied to complex models like CLM. A full description of the DREAM$_{(zs)}$ algorithms can be found in (Vrugt, 2015).

To constrain the C3-crop parameters, a one-year time series of non-gap filled, half-hourly NEE data from the EC tower at the ME site (1 Dec 2011 – 30 Nov. 2012) was used. The five key parameters are: (1) the fraction of leaf N in Rubisco enzyme (fl$_{NR}$), (2) the growth respiration factor ($g_R$), (3) the rooting distribution parameter [1/m] ($r_b$), (4) the specific Leaf Area at top of canopy [m$^2$/gC] (sla$_{top}$), (5) the soil water potential at full stomatal closure [mm] ($\psi_c$). This selection is based on Post et al. (2016) and previous studies , which highlighted the sensitivity of these parameters (Foereid et al., 2014; Göhler et al., 2013).

Parameter estimates for C3-grass, evergreen coniferous forest and broadleaf deciduous forest were already successfully estimated and validated for central European sites by Post et al. (2016) with DREAM$_{(zs)}$-CLM4.5 and therefore adopted in this study. Post et al. (2016) used NEE data from the RO and WÜ site to estimate parameters for C3-grass and coniferous forest, as well as FLUXNET data from site Fontainebleau (FR-Fon) in France, located about 300 km south-west of the Rur catchment [48.4763 N, 2.7801 E] (e.g., Migliavacca et al., 2015). The validation was based on NEE data from EC sites of corresponding PFTs that were located ~ 600 km away from the parameter estimation sites. Post et al. (2016) estimated the parameters fl$_{NR}$, $g_R$, $r_b$, sla$_{top}$, and $\psi_c$ jointly with three more parameters: the temperature coefficient ($Q_{10}$), the base rate for maintenance respiration ($mr_b$) and the Ball-Berry slope of conductance-photosynthesis relationship ($b_s$). $Q_{10}$ quantifies the fractional change of the respiration rate in response to a 10°C temperature rise. $Q_{10}$, $mr_b$ and $b_s$ in CLM4.5 are by default hard-wired in the CLM4.5 source code and not defined separately for each PFT like the other five parameters. This implies that one single value of those parameters is defined globally, which is then applied to all PFTs. However, Post et al. (2016) found that also values of the estimated hard-wired parameters, especially $Q_{10}$, varied for different PFTs or sites. The fact that $Q_{10}$, $mr_b$ and $b_s$ are hard-wired in CLM imposes a challenge when applying the jointly estimated parameter sets on regional scale, because it requires that $Q_{10}$, $mr_b$ and $b_s$ can be defined separately for each PFT. Therefore, we modified CLM in order to input the PFT-specific values for $Q_{10}$, $mr_b$ and $b_s$ of the jointly estimated parameter sets by Post et al. (2016). Post et al. (2016) showed that the estimated parameter values for C3-crop did not enhance model performance compared to the default parameter values, if applied to an evaluation year or another C3-crop site. This applies to the parameter values that had been estimated and applied using a one-year time series of NEE data, as it was done in this study. Therefore, we estimated a new set of C3-crop parameters herein.

### 2.3.2. Evaluation of parameter estimates with eddy covariance NEE data

In order to evaluate the performance of the estimated CLM C3-crop specific parameters in terms of the consistency of modelled and measured data, two CLM cases were defined and compared: (i) A reference run (CLM-Ref): a forward run for the years 2011-2013 with default parameters and the atmospheric input data and initial conditions as described in Sect. 2.2.; (ii) An ensemble run with 60 realizations, for the same time period, same initial conditions and same atmospheric input data as CLM-Ref, but with parameters sampled randomly from the DREAM multivariate posterior pdfs (CLM-Ens$_P$). This

implies that we did not perform a separate model spin-up for each of the estimated set of parameters, as this was computationally too expensive. The parameter estimates for ME were evaluated with eddy covariance NEE data from four different C3-crop sites in the Rur catchment (ME, SE, NZ and EN). In case of ME, the NEE observation time series of the year that followed the parameter estimation year were used for evaluation (Table 1).

The evaluation was conducted with time series of half hourly NEE data measured at the EC tower sites. First it was verified that the PFT coverage of one CLM grid cell coincided with the dominant PFT at the respective EC tower site. Each of the four grid cells was covered more than 80% by C3-crop. This was considered sufficient to allow for a grid-based model evaluation. In the following, the subscript "gc" is used to refer to the grid cell, in which one of the EC towers is located, e.g. "$ME_{gc}$" to the respective grid cell of the ME site. The length of the available NEE time series differed among the EC sites,

ranging from two to twelve months (including data gaps). See Figure 1. Accordingly, only the model output that coincides with the observed NEE data was used.

The calculated NEE output was evaluated based on the following indices:

The root mean square error ($RMSE_m$):

$$RMSE_m = \sqrt{\frac{1}{n}\sum_{i=1}^{n}(m_i - y_i)^2},$$ (1)

with $y$ = measured half hourly NEE [$\mu$mol m$^{-2}$ s$^{-1}$] at time step $i$ for the given time series of length $n$, and $m$ = modelled

equivalent.

The mean absolute difference of the mean diurnal NEE cycle ($MAD_{dir}$):

$$MAD_{dir} = \frac{1}{48}\sum_{i=1}^{48}|\overline{m_d} - \overline{y_d}|,$$ (2)

with $\bar{y}$ = average measured NEE at a given time $d$ during the day [$\mu$mol m$^{-2}$ s$^{-1}$] and $\bar{m}$ = modelled equivalent. This performance measure is evaluated half hourly and therefore 48 times per day. For the sites ME and SE, where a complete year of NEE data was available, the mean diurnal NEE cycle and the index $MAD_{dir}$ were calculated separately for each of

20 the four seasons within the evaluation year (winter: Dec.-Feb.; spring: Mar.-May; summer: Jun.-Aug.; autumn: Sept.-Nov.). In order to summarize the four seasonal $MAD_{diur}$ indices to one $MAD_{diur}$ index representative for the whole evaluation year, the four seasonal indices were averaged. For the other sites where only NEE data for ~ 2-3 months were available, the indices including $MAD_{dir}$ were calculated for this shorter time series.

The relative difference [%] of the NEE sum ($\sum$NEE) calculated for all half hourly data available in the respective evaluation

period ($RD_{\sum NEE}$):

$$\text{RD}_{\sum\text{NEE}} = \frac{\sum_{i=1}^{n}(m_i) - \sum_{i=1}^{n}(y_i)}{\sum_{i=1}^{n}(y_i)} \times 100, \tag{3}$$

with $y$ = measured NEE (non-gap filled), and $m$ = modelled equivalent.

All evaluation indices were determined both for CLM-Ref and for the ensemble mean of CLM-Ens$_P$. Post et al. (2016) showed that parameters estimated for the forest sites WÜ and FR-Fon are well transferable to other FLUXNET sites of corresponding PFTs (Tharandt (Germany) (DE-Tha) (e.g., Grünwald and Bernhofer, 2007), Hainich (Germany) (DE-Hai) (e.g., Knohl et al., 2004)) located in more than 600 km distance from WÜ and FR-Fon. Because WÜ is the only forested EC tower site in the Rur-catchment and no additional EC towers were available for the validation of the estimated parameters, we assume the parameter estimates provide more reliable NEE data for a large part of the forested area in the Rur catchment. We also adopted successfully estimated and validated parameters for the C3-grass site RO from Post et al. (2016).

### 2.3.3. Evaluation of LAI predictions

Various studies highlight current deficits to accurately estimate and validate LAI derived from multispectral satellite images for deciduous and needleleaf forest (Ganguly et al. 2012; Tillack et al. 2014; Härkönen et al. 2015). Also the LAI data derived from RapidEye (LAI$_{\text{RapidEye}}$) used herein have not been validated yet for the forest PFTs. Thus, the model performance in terms of the LAI was only evaluated for grid cells which were covered by more than 80% by C3-grass or C3-crop, not for the forest PFTs. The effect of the parameter estimates on LAI was evaluated for the 1 km$^2$ grid of the Rur catchment domain. LAI$_{\text{RapidEye}}$ data of about 18 days (depending on the location) between the 1 Nov 2011 and the 16 Sep 2012 were used for the LAI evaluation and compared with modelled LAI data on those days and those grid cells for which RapidEye data were available. The LAI was evaluated separately for C3-grass and C3-crop, and both for the winter half year (Nov.-Apr.) and summer half year (May-Oct.). To evaluate and compare modelled LAI for CLM-Ens$_P$ and CLM-Ref, the mean absolute difference between simulated LAI and measured LAI (MAD$_{\text{LAI}}$) was calculated over all grid cells that were covered by >80 % with C3-grass ($n_{\text{pft}}$ = 224) or C3-crop ($n_{\text{pft}}$ = 404):

$$\text{MAD}_{\text{LAI}} = \frac{1}{n_{\text{days}} * n_{\text{pft}}} \sum_{i=1}^{n_{\text{days}}*n_{\text{pft}}} |m_i - y_i| \tag{4}$$

with $m$ = modelled daily LAI [m$^2$ m$^{-2}$ day$^{-1}$] and $y$ = measured equivalent, $n_{\text{days}}$ = number of days, and $n_{\text{pft}}$ = number of grid cells for which LAI$_{\text{RapidEye}}$ data were available at a particular day. RMSE$_{\text{LAI}}$ was calculated according to Eq. (1).

The mean LAI for each PFT was calculated by:

$$\text{LAI}_{\text{PFT}} = \frac{1}{n_{\text{days}} * n_{\text{pft}}} \sum_{i=1}^{n_{\text{days}}*n_{\text{pft}}} \text{LAI}_i \tag{5}$$

with $n_{\text{days}}$= number of days RapidEye data for a given PFT were available, and $\text{LAI}_i$= LAI observed or modelled at a particular day [$m^2\ m^{-2}$ day$^{-1}$]. For each $\text{LAI}_{\text{PFT}}$ value, a respective standard deviation was calculated.

## 2.4. Uncertainty estimation

### 2.4.1. Perturbation of atmospheric input data

In order to take the uncertainty of the meteorological input data into account, a 60 member ensemble of perturbed meteorological forcings was generated for the years 2008-2012 using hourly COSMO-DE data. The approach used to generate the perturbation fields has previously been applied in soil moisture data assimilation studies (Reichle et al., 2007, 2010; Kumar et al., 2012; Han et al., 2012, 2013, 2014). Perturbation fields were applied to air temperature [K], incoming long wave radiation [W/m$^2$], incoming short wave radiation [W/m$^2$] and precipitation [mm/s]. Normally distributed additive perturbations were applied to longwave radiation (LW) and air temperature (Temp). Log-normally distributed multiplicative perturbations were applied to precipitation (Prec) and shortwave radiation (SW). The parameters used for the perturbations were adapted from Han et al. (2014) and are listed in Table 2. The perturbations for the different atmospheric variables are cross-correlated in order to generate physically plausible perturbations of the atmospheric forcings. We considered spatial correlation in addition to temporal correlation of the meteorological variables. The multiplicative perturbations are truncated by a defined maximum of 2.5 standard deviations, to remove outliers from the generated perturbation fields (Reichle et al., 2007). The spatially correlated noise is calculated first using the Fast Fourier Transform approach (Park and Xu, 2013) with a 10 km spatial correlation scale. Next, the temporally correlated noise is added to the spatially correlated noise. The temporal correlation for all perturbed variables is imposed using a first-order AR(1) autoregressive model (Reichle et al., 2007). The AR(1) temporal correlation coefficient for a time lag of one day was 0.368 for all variables.

### 2.4.2. Generation of perturbed initial state input files and the perturbed forward run for the Rur catchment

As shown in various studies, carbon fluxes predicted by land surface models strongly depend on the carbon-nitrogen pools generated during the model spin-up (Carvalhais et al., 2010; Kuppel et al., 2012). In order to take the uncertainty of initial states into account, a 60 member ensemble of perturbed initial states was generated. This was done via a 15 year spin-up using perturbed atmospheric forcings for the years 2008-2010 (Sect. 2.4.1) and parameter values sampled randomly from the joint DREAM posterior pdf's. The initial conditions used at the beginning of the 15 year perturbed spin-up were taken from the end of the main 1200 years spin-up plus three years exit spin-up. This perturbation takes into account the uncertainty of the less stable carbon and nitrogen pools, but uncertainty with respect to the stable and resistant carbon and nitrogen pools is

not taken into account. As the stable nitrogen pool is not perturbed, the availability of N from mineralization is very similar across the ensemble members. This might be an unproblematic assumption given N fertilization by agriculture and atmospheric deposition, but also implies that the uncertainty of the net N availability at the sites is likely underestimated.

### 2.4.3. Evaluation of model uncertainty

In order to evaluate the effect of uncertain CLM parameters, meteorological input data and initial model states, we setup three CLM ensembles in addition to CLM-Ens$_P$ with 60 ensemble members each:

(i)      CLM-Ens$_P$: ensemble model runs for 1 Dec 2012 – 30 Nov 2013 with deterministic (non-perturbed) initial states and non-perturbed input data from COSMO-DE (Sect.2.2). The initial states are the outcome of the default 1203 years spin-up, without perturbations. The parameter values were sampled randomly from the DREAM$_{(zs)}$

multivariate posterior pdfs (Table 3), which were estimated for C3-crop or adopted from Post et al. (2016) in case C3-grass, evergreen coniferous forest and broadleaf deciduous forest. In case of C3-crop, the setup of CLM-Ens$_P$ was identical to the one used for the evaluation of the estimated C3-crop parameters in terms of the NEE model-data consistency (Sect. 2.3.2).

(ii)      CLM-Ens$_{PA}$: ensemble model runs for 1 Dec 2012 – 30 Nov 2013 with parameter values sampled according to

CLM-Ens$_P$ and perturbed atmospheric forcings, but using deterministic initial states.

(iii)      CLM-Ens$_{PAI}$: ensemble model runs according to CLM-Ens$_{PA}$ but starting from perturbed initial states.

(iv)      CLM-Ens$_{P+Q10}$: a forward run for the period 1 Dec 2012 – 30 Nov 2013 with deterministic forcings, deterministic initial states and parameters sampled according to CLM-Ens$_P$, except for $Q_{10}$. $Q_{10}$ values for C3-crop and forest PFTs were sampled from a normal distribution with a mean of 2.0 and a standard deviation of 0.5 with a minimum

and maximum bound set to 1.3 and 3.0. We choose $Q_{10}$ because of its central role for carbon stock and flux predictions and the respective uncertainties in most land surface models (Post et al., 2008; Hararuk et al., 2014), including CLM (Post et al. 2016). The objective of ensemble CLM-Ens$_{P+Q10}$ was to estimate the uncertainty of the CLM output induced by the prior uncertainty of one single key parameter, and compare it to the posterior uncertainty from CLM-Ens$_p$.

The modelled NEE sum [gC m$^{-2}$ y$^{-1}$] for the period 1 Dec 2012- 30 Nov 2013 was calculated for each ensemble member using the complete time series of daily outputs. The PFT-specific NEE sum ($\sum$NEE$_{PFT}$) was then calculated separately for each of the four main PFTs by averaging the simulated NEE sum over all grid cells that were covered by more than 80% of the particular PFT. The analysis of model uncertainty was then based on the standard deviation ($\sigma$) of $\sum$NEE$_{PFT}$ between the different ensemble members, $\sigma(\sum$NEE$_{PFT})$. Accordingly for GPP and ER, $\sigma(\sum$GPP$_{PFT})$ and $\sigma(\sum$ER$_{PFT})$ were calculated.

# 3. Results

## 3.1. Evaluation of simulated NEE and LAI with estimated parameter values

As shown in Table 4, CLM parameter estimates notably reduced the mismatch between modelled and measured NEE data in comparison with the reference run (Ref), where global default parameter values were used.

MAD$_{diur}$ was reduced by 7% (SE$_{gc}$), 25% (NZ$_{gc}$) and 35% (ME$_{gc}$) and increased by 7% for EN$_{gc}$. Thus, the mean diurnal NEE cycles were closer to the observed ones for all evaluation grid cells dominated by C3-crop, except EN$_{gc}$, if estimated parameters were used. The RMSE$_{m}$ was reduced by up to 13% (SE$_{gc}$) for all sites except NZ$_{gc}$. RD$_{\sum NEE}$ was most notably reduced compared to the other evaluation indices. The measured $\sum$NEE was negative for each of the four EC sites and for all of the respective model outputs (Figure 2). This would imply that all sites were net carbon sinks during the evaluation period. However, because data gaps were included in the NEE time series and because more EC data were available for summer and daytime than for winter and nighttime, those values do not represent the true NEE sum of the evaluation period. For CLM-Ref and all evaluation sites, $\sum$NEE differed notably from the observed data (Figure 2, Table 4). For ME$_{gc}$, SE$_{gc}$ and NZ$_{gc}$, the predicted $\sum$NEE was significantly closer to the observations for CLM-Ens$_{P}$ than for CLM-Ref, and RD$_{\sum EE}$ was reduced by a factor 1.2 (SE$_{gc}$) to 7.2 (ME$_{gc}$). RD$_{\sum EE}$ was 13% (SE$_{gc}$) to 68% (ME$_{gc}$) lower for CLM-Ens$_{P}$ compared to CLM-Ref. For EN$_{gc}$, RD$_{\sum EE}$ was slightly lower for CLM-Ref, but as indicated in Figure 2, the difference of $\sum$NEE was not significant. Overall, results indicate that NEE sums over time period from several months to one year were better represented with estimated values than with global default values. Figure 2 highlights that the uncertainty of modelled NEE is probably underestimated, if the uncertainty of meteorological input data and initial states is not taken into account. The standard deviation of CLM-Ens$_{PAI}$ is notably higher compared to CLM-Ens$_{P}$, where only parameter uncertainty is considered.

Table 5 summarizes the catchment average leaf area indices and the corresponding mean absolute differences (MAD$_{LAI}$) and RMSE between the observed and modelled LAI$_{PFT}$. In the summer half year, LAI$_{Ens}$ was closer to LAI$_{RapidEye}$ than LAI$_{Ref}$, both, for C3-grass and C3-crop. RMSE$_{LAI}$ was on average 1.5 for CLM-Ens$_{P}$ and 3.4 for CLM-Ref. MAD$_{LAI}$ for summer was 1.2 for CLM-Ens$_{P}$ and 3.4 for CLM-Ref on average. In winter, the difference was only 0.2 (RMSE$_{LAI}$) and 0.3 (MAD$_{LAI}$) and thus the improvement not significant.

While CLM-Ref overestimated LAI$_{PFT}$, ENS$_{P}$ underestimated LAI$_{PFT}$. Overall, the standard deviation of LAI$_{PFT}$, i.e. the LAI variation over all included grid cells *n* throughout the half year period was very high, both for the observed and the modelled data. The mean annual LAI cycles for ME$_{gc}$ and SE$_{gc}$ (Figure 3) indicate that the uncertainty of LAI was underestimated for CLM-ENS$_{P}$. The uncertainty of simulated LAI was much higher for CLM-Ens$_{PAI}$.

Figure 3 highlights that in case of C3-crop, the simulated yearly LAI cycle did not match well the observed and expected annual LAI course. The delay of the plant emergence indicated by these LAI courses is related to the strong underestimation of daytime NEE in spring 2013 (Figure 4, Figure 5). This underestimation of NEE can mainly be attributed to an

underestimation of GPP, which is too low in spring because the simulated plant onset was about 2 weeks later than observed. For CLM-Ref and for most of the CLM-Ens$_{PAI}$ ensemble members, leaf onset started in May. For those model realizations, the underestimation of daytime NEE in spring was highest. In contrast, for CLM-Ens$_P$ and a small proportion of CLM-Ens$_{PAI}$, leaf onset already started in March. For those cases, the underestimation of daytime NEE in spring was notably

lower. This elucidates the close link of modelled NEE and LAI and highlights that errors in the timing of leaf onset can lead to substantial errors in simulated NEE. The evaluation of simulated LAI showed that modelled and observed LAI are closer for simulations with estimated CLM parameters than for simulations with default parameters and that in case of C3-crop, simulated leaf onset was better represented.

### 3.2.      Uncertainties of simulated carbon fluxes on catchment scale

In this section, results of modelled NEE, GPP and ER are summarized and compared for the CLM ensembles CLM-Ens$_P$, CLM-Ens$_{PA}$, CLM-Ens$_{PAI}$ and CLM-Ens$_{P+Q10}$ with focus on the model uncertainty. The uncertainty is evaluated with the standard deviations $\sigma(\sum NEE_{PFT})$, $\sigma(\sum GPP_{PFT})$ and $\sigma(\sum ER_{PFT})$ of the 60 ensemble members. In this case the time series from 1 Dec 2012 to 30 Nov 2013 of the simulated fluxes was used to calculate the PFT-specific, catchment average sums $\sum NEE_{PFT}$, $\sum GPP_{PFT}$ and $\sum ER_{PFT}$ [gC m$^{-2}$ s$^{-1}$].

Figure 6 shows the means and standard deviations of $\sum NEE_{PFT}$ for the four CLM ensembles and for CLM-Ref. The sign of $\sum NEE_{PFT}$ indicates that for CLM-Ref, all PFTs in the catchment act as net carbon sources. $\sum NEE_{PFT}$ changed significantly if estimated parameter values were applied. In that case, all PFTs except C3-grass acted as carbon sink and source. Figure 7 illustrates how estimated parameters affected the modelled annual NEE sum [gC m$^{-2}$ y$^{-1}$] for the Rur catchment. With default parameter values, the annual NEE was positive for most of the grid cells, particular in the northern part of the catchment

which is predominated by agriculture. In terms of NEE, the catchment would be a net $CO_2$ source. With estimated parameters from CLM-Ens$_P$, the NEE sum became negative for most grid cells in the catchment including a large part of the northern lowland area (Figure 7). Thus, with estimated parameters the catchment became a net $CO_2$ sink (disregarding $CO_2$ fluxes due to harvesting, land use change or fossil fuel combustion). Figure 7 indicates that both GPP and ER increased with estimated parameters. However, GPP increased more than ER. Since we found that at verification sites simulations with

estimated parameters gave NEE sums closer to the observed NEE sums than simulations with default parameters, we assume the catchment scale NEE sum (Figure 7) and $\sum NEE_{PFT}$ (Figure 6) are more reliable for CLM-Ens$_P$ than for CLM-Ref.

As indicated in Figure 6, for C3-grass and C3-crop, $\sigma(\sum NEE_{PFT})$ was significantly higher for CLM-Ens$_{PA}$ compared to CLM-Ens$_P$. For CLM-Ens$_{PA}$, $\sigma(\sum NEE_{PFT})$ was ~45 gC m$^{-2}$ y$^{-1}$ (C3-grass ) and ~75 gC m$^{-2}$ y$^{-1}$ (C3-crop) compared to ~2-3 gC m$^{-2}$ y$^{-1}$ for CLM-Ens$_P$. Thus, applying perturbed forcings led to a very strong increase of the uncertainty of simulated $\sum NEE_{PFT}$ by

a factor of ~14 (C3-crass) and ~42 (C3-crop). This finding reveals that C3-grass and C3-crop in CLM were extremely sensitive to minor differences in the meteorological input data. In comparison, the differences of $\sigma(\sum NEE_{PFT})$ between CLM-

Ens$_{PA}$ and CLM-Ens$_{PAI}$ were minor. Thus, also the much higher $\sigma(\sum NEE_{PFT})$ for CLM-Ens$_{PAI}$ compared to CLM-Ens$_P$ can mainly be ascribed to the perturbed meteorological input data, rather than the perturbed initial states. Due to the large uncertainty of $\sum NEE_{PFT}$ for both CLM-Ens$_{PA}$ and CLM-Ens$_{PAI}$, differences of the ensemble mean $\sum NEE_{PFT}$ were not significant between the different two CLM-ensembles. For C3-crop and C3-grass, $\sum GPP_{PFT}$ and $\sum ER_{PFT}$ were strongly sensitive to the perturbed forcings such that the strong increase of $\sigma(\sum NEE_{PFT})$ for CLM-Ens$_{PA}$ compared to CLM-Ens$_P$ is related to changes in both ER and GPP (Figure 6).

For coniferous forest and deciduous forest, $\sigma(\sum NEE_{PFT})$ for CLM-Ens$_{PA}$ and CLM-Ens$_{PAI}$ were ~ 4.0 -13.5 gC m$^{-2}$ y$^{-1}$ and thus considerably lower in comparison to C3-grass and C3-crop. The increase of $\sigma(\sum GPP_{PFT})$ and $\sigma(\sum ER_{PFT})$ with perturbed forcings and perturbed initial states was also minor in comparison to CLM-Ens$_P$. This indicates that the carbon cycle of the forest PFTs in CLM is much less sensitive to the atmospheric input data compared to C3-grass and C3-crop. However, when comparing CLM-Ens$_{PA}$ and CLM-Ens$_{PAI}$, the additional effect of the perturbed initial states was notably higher for forest compared to C3-grass and C3-crop. For the forest PFTs, the ensemble mean $\sum NEE_{PFT}$ changed significantly if initial states were perturbed (Figure 6). This was related to the fact that estimated parameters and perturbed atmospheric forcings disrupted the steady state of the forest carbon pools. Hence, carbon pools increased relatively rapidly during the perturbed spin-up (Sect. 2.4.2), which was not the case for C3-grass and C3-crop. As already shown in the previous section, the spread of CLM-Ens$_P$ was low, indicating that the uncertainty of simulated NEE induced by the posterior parameter values that were sampled from the estimated pdfs was low. This is however related to the fact that parameters were already conditioned to NEE data, which reduced the parameter uncertainty. The additional perturbation of $Q_{10}$ (CLM-Ens$_{P+Q10}$) increased $\sigma(\sum NEE_{PFT})$ by a factor of ~6 (C3-grass, coniferous forest) to ~19 (deciduous forest) in comparison to CLM-Ens$_P$. This highlights the strong effect of an uncertain $Q_{10}$ parameter on the uncertainty of predicted carbon fluxes in CLM. As shown in Figure 6, the perturbed $Q_{10}$ parameter strongly affected both $\sum ER_{PFT}$ and $\sum GPP_{PFT}$. For the forest PFTs, the effect of $Q_{10}$ (Ens$_{P+Q10}$) on the uncertainty of the regional scale $\sum GPP_{PFT}$, $\sum ER_{PFT}$ and $\sum NEE_{PFT}$ was much stronger than the effect of both uncertain meteorological forcings and initial states (CLM-Ens$_{PAI}$).The ensemble mean $\sum ER_{PFT}$ did not change significantly, except for C3-grass. However, for coniferous forest, the ensemble means for CLM-Ens$_{P+Q10}$ of $\sum ER_{PFT}$ and $\sum NEE_{PFT}$ were significantly different from CLM-Ens$_P$. This is not surprising, since the mean of $Q_{10}$=2.0 in case of CLM-Ens$_{P+Q10}$ was considerably lower than the $Q_{10}$ values of CLM-Ens$_P$ sampled from the estimated parameter sets (Table 3).

## 4.    Discussion

The results of the study showed that the model-data consistency was enhanced with estimated parameters for RO$_{gc}$, ME$_{gc}$, WÜ$_{gc}$, SE$_{gc}$ and NZ$_{gc}$. For KA$_{gc}$ and EN$_{gc}$, not all evaluation indices indicated an improvement of modelled NEE or the improvement was not significant. For the roving station sites NZ, KA and EN, NEE time series of only 2-3 months were available for evaluation in contrast to the other sites, where time series of a whole year were available. Accordingly,

evaluation results were not directly comparable between those sites. Post et al. (2016) showed that the evaluation runs with parameter estimates had a strongly varying performance over the year. Thus, if a complete year of NEE data had been available for evaluation of the roving station sites, evaluation results for those sites would have been more informative. Particularly the transfer of parameters estimated for C3-grass and C3-crop to the catchment domain was found critical. As

expected, parameter values estimated for the winter wheat site ME did not enhance the model performance at the grid cell $EN_{gc}$ which was predominated by sugar beet. Accordingly, results indicated that parameter values estimated for one C3-crop site, are not necessarily transferable to other C3-crop types characterized by a different physiology and management. This highlights the limitations of large scale modeling with land surface models that only distinguish between very broad groups of PFTs. For $SE_{gc}$, where also winter wheat was grown during the evaluation period like in ME, parameter estimates clearly

improved the model performance in terms of NEE. Several studies already emphasized that LSM parameters can vary within one group of PFT such that the transfer from single site estimates to other sites with the same PFT is not trivial (Groenendijk et al., 2011; Xiao et al., 2011). In addition to deficits in the representation of plant phenology, management (harvesting, cutting, etc.) is not explicitly considered for C3-crop and C3-grass in CLM although it has a significant impact on NEE (Borchard et al., 2015). Therefore, observed temporal LAI variations in the growing season were not well represented in the

default model simulations. However, estimated parameters by CLM-$Ens_P$ provided more reliable estimates of the NEE sums for ME and SE than the default run because the modelled plant emergence was shifted ahead. Therefore, we expect that the performance of CLM can be improved if it is coupled to a crop model, which can treat specific plant traits and include management practices. Li et al. (2011) already found for the coupled LSM-crop model ORCHIDEE-STICS that crop variety and management factors like irrigation, fertilization and planting date have a significant impact on NEE (and

evapotranspiration). Wu et al. (2016) showed that a crop model coupled to the LSM ORCHIDEE, combined with assimilation of many different data types, was able to reproduce many measurement data up to the measurement uncertainty, whereas other remaining errors could be related to management activities which were not correctly represented in the model.

For the forest PFTs, the uncertainty of the $Q_{10}$-parameter had the largest impact on NEE, followed by uncertainty of the initial states and uncertainty of meteorological forcings. The crucial role of the $Q_{10}$ parameter for carbon stock and flux

predictions and the respective uncertainties in most land surface models was already highlighted in previous studies (Post et al., 2008; Hararuk et al., 2014, Post et al., 2016). Post et al. (2016) showed that $Q_{10}$ correlates strongly with other CLM4.5 key parameters like $fl_{NR}$ and the Ball-Berry slope of stomatal conductance as well as with the initial carbon-nitrogen pools. This may explain why the spread of the regional scale $\sum GPP_{PFT,} \sum ER_{PFT}$ and $\sum NEE_{PFT}$ in $Ens_{P+Q10}$ was notably higher for coniferous forest than for the other PFTs (Figure 6).

It is important to stress again that uncertainty in initial states was considered by an additional 15 years spin-up with perturbed parameters and meteorological forcings. This short additional spin-up does not consider uncertainty of the most stable carbon and nitrogen pools. Carbon and nitrogen pools are also influenced by changing land use in the last centuries and management practices, which are also not considered. Therefore, the true uncertainty related to the initial states will be

larger than in these simulation experiments. The simulation results for C3-grass and C3-crop were surprising as uncertainty in the meteorological forcings had a much larger impact on uncertainty of NEE than the uncertainty of initial states and $Q_{10}$ had. We argue that the applied meteorological perturbations (standard deviations of 1.0K for air temperature, 30% of the incoming shortwave radiation and 50% of precipitation amount) are realistic for meteorological reanalysis products, and in correspondence with other studies. The perturbations (which are also temporally correlated) result in considerable variations, between ensemble members, of leaf onset and senescence. The stress deciduous phenology scheme (which determines plant onset and offset for C3-crop and C3-grass in CLM as outlined in Section 2.2) is strongly determined by various arbitrary thresholds such as "crit_offset_swi" (water stress days for offset trigger) or "crit_onset_fdd" (critical number of freezing degree days to trigger onset). The phenology representation of C3-crop and C3-grass could be too sensitive to those thresholds (Dahlin et al. 2015). Verheijen et al. (2015) argue that the use of PFTs instead of plant traits in LSMs reduces the adaptive response of vegetation to environmental drivers.

A second possible reason for the large impact of the perturbation of meteorological forcings on NEE-uncertainty is the preferential location of the C3-crop and C3-grass in the drier, northern part of the catchment, which sometimes experiences drought stress during summer. The perturbation of the precipitation amounts affects drought stress and can enhance the variability in simulated NEE and LAI for C3-crops and C3-grasses in drier areas. Thus, the impact of the uncertainty of meteorological forcings depends on the local conditions. Jung et al. (2011) already showed that interannual variability of NEE is larger in semi-arid and semi-humid areas than in other regions. In summary, the large impact of uncertainty of meteorological input on NEE-uncertainty is likely primarily model specific, but also related to the semi-humid conditions in the agricultural part of the region. High impact of uncertainty of meteorological conditions can be expected in other semi-humid and semi-arid regions and should be taken into account in simulation studies.

In order to better estimate the uncertainty of the initial carbon and nitrogen pools, it would be necessary to include the spin-up (for each of the ensemble members) in the data assimilation experiments where parameters are estimated. As model runs have to be repeated thousands of times, this would be extremely CPU-expensive. The spin-up should also consider uncertainty of historical land use and uncertainty of historical meteorological conditions. Especially the land use and meteorological conditions of the last few centuries would impact the initial states. We believe that it is important to consider jointly the uncertainty of initial conditions and parameters in LSM forward and inverse modelling runs. Pinnington et al. (2016) already pointed towards the importance of considering correlations between initial states and parameters in inverse modelling studies.

## 5.    Conclusions

This study evaluated ecosystem parameters estimated with DREAM$_{(zs)}$ for the regional scale land surface model CLM4.5. The ecosystem parameters were estimated for EC sites with different plant functional types (PFTs) in western Germany and

northern France by conditioning to time series of NEE data. The estimated parameters were assigned to a distributed high resolution land surface model for the Rur catchment in Germany. It was evaluated whether the distributed land surface model reproduced measured NEE and LAI better with estimated parameters than with default parameters. Moreover, a comprehensive model uncertainty analysis was done for the four main PFTs in the catchment, comparing the effect of different model error sources (parameters, atmospheric forcings and initial states) on the CLM ensemble spread.

Parameter estimation using NEE data was found suitable to improve LAI predictions. In case of C3-crop, the timing of plant emergence in spring was more accurate. This resulted in an improved characterization of NEE, which was highly underestimated in spring in cases where the plant emergence was delayed. This highlights the potential of DREAM-CLM parameter estimates to utilize individual observations at few grid cells to improve catchment wide predictions of LAI and carbon fluxes.

Estimated parameters reduced the relative difference between the observed and modelled NEE sum significantly for most evaluation sites compared to a reference run with default parameters. For all PFTs except C3-grass, the sign of the mean annual NEE sum over the catchment was reversed from positive to negative with estimated parameters compared to default parameters. This implies that forest and C3-crop areas in catchment were predicted as net $CO_2$ sources with global default parameter values and as net $CO_2$ sinks with the successfully validated parameter estimates. This elucidates the potential and relevance of parameter estimation in terms of obtaining more reliable estimates of regional carbon balances.

The uncertainty of predicted NEE and LAI was underestimated, if only estimated parameters were sampled without additional consideration of uncertain atmospheric input data and uncertain initial conditions. Constraining CLM parameters with DREAM$_{(zs)}$ resulted in a very low uncertainty of the predicted NEE and LAI. However, with additional consideration of uncertainty in the initial model conditions and atmospheric input data, the uncertainty of NEE was significantly higher. Thus, taking into account the uncertainty of parameters, atmospheric input data and initial states for predictions of carbon fluxes or stocks with CLM4.5 is considered essential.

The effect of uncertainty of atmospheric forcing data on the LAI and NEE uncertainty was considerably higher for C3-grass and C3-crop than for the forest PFTs. This is related to the C3-crop/C3-grass specific phenology representation and model internal thresholds in CLM. Many of these thresholds relate to temperature and strongly control leaf onset and senescence. The strong effect of the perturbed forcing data on modelled carbon fluxes in CLM4.5 is closely linked to the effect of uncertain model parameters like the temperature coefficient $Q_{10}$.

The model uncertainty resulting from uncertain atmospheric forcing data for C3-grass and C3-crop is additionally enhanced because these crops are located in the northern part of the catchment which is prone to some drought stress in summer. In combination with soil moisture related model thresholds, the effect of the perturbed precipitation on modelled NEE and LAI can be regionally different due to differences in the regional climate. This stresses the importance of regional scale modelling.

This study highlighted that if different crop types are grown in a region, it is particularly important to parameterize those crop types independently in order to obtain reliable carbon flux estimates with a land surface model like CLM. A separate differentiation of different crop types including a crop-specific representation of phenology and management is of high importance to obtain more reliable carbon flux and stock estimates in the future, as it is already foreseen for future CLM versions (personal communication with collaborators from the CLM development team at NCAR).

## Acknowledgments

This work was carried out with the funding of the Transregional Collaborative Research Centre 32 (TR32) and the EU FP7 project ExpeER (Grant Agreement no. 262060) that is supported by the European Commission through the Seventh Framework Programme for Research and Technical Development. We gratefully acknowledge the TERENO and the Transregional Collaborative Research Centre 32 (TR32), for providing eddy covariance and additional data for the sites Rollesbroich, Merzenhausen, Selhausen, Niederzier, Kall and Wüstebach. For allocating the FLUXNET data (Fr-Fon) used in this study, we are exceptionally thankful to Eric Dufrene. The authors gratefully acknowledge the computing time granted on the supercomputer JUROPA at the Jülich Supercomputing Centre (JSC). RapidEye data was provided by Blackbridge with the RapidEye Science Archive (RESA). We also thank Pramod Kumbhar and Guowei He for their support on high performance computing.

**Appendix**

**A1. The DREAM$_{(zs)}$ parameter estimation approach**

The adaptive Markov Chain Monte Carlo (MCMC) method DREAM$_{(zs)}$ (Ter Braak and Vrugt, 2008; Laloy and Vrugt, 2012; Vrugt, 2015) estimates the posterior pdf of model parameters based on Bayes theorem:

$$p(\mathbf{x}|\widetilde{\mathbf{Y}}) = \frac{p(\mathbf{x})p(\widetilde{\mathbf{Y}}|\mathbf{x})}{p(\widetilde{\mathbf{Y}})}$$

5    where $\mathbf{x}$ are the model parameters to be estimated, $\widetilde{\mathbf{Y}} = \{\tilde{y}_1, \ldots, \tilde{y}_n\}$ is a $n$-vector of measured data, $p(\mathbf{x}|\widetilde{\mathbf{Y}})$ is the posterior probability density function (pdf), $L(\mathbf{x}|\widetilde{\mathbf{Y}}) \equiv p(\widetilde{\mathbf{Y}}|\mathbf{x})$ the likelihood function, $p(\mathbf{x})$ the prior distribution and $p(\widetilde{\mathbf{Y}})$ the normalizing constant. In practice, statistical inferences about $p(\mathbf{x}|\widetilde{\mathbf{Y}})$ are made from its unnormalized density, $p(\mathbf{x}|\widetilde{\mathbf{Y}}) \propto p(\mathbf{x})L(\mathbf{x}|\widetilde{\mathbf{Y}})$.

Whether a proposal point of chain $i$ at iteration $t$ is accepted and thus

$$\mathbf{X}_p^i = \mathbf{X}^i + d\mathbf{X}^i$$

10    is determined with the Metropolis acceptance ratio:

$$P_{\text{accept}}\left(\mathbf{x}_{t-1}^i \to \mathbf{X}_p^i\right) = \min\left[1, \frac{p(\mathbf{X}_p^i)}{p(\mathbf{x}_{t-1}^i)}\right].$$

If the candidate point is accepted, then the $i^{\text{th}}$ chain moves to the new position, that is $\mathbf{x}_{t-1}^i = \mathbf{X}_p^i$, otherwise $\mathbf{x}_t^i = \mathbf{x}_{t-1}^i$ (Vrugt, 2015).

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

Tables

**Table 1. Eddy covariance tower sites in the Rur catchment.**

| EC site name | Latitude [°N] | Longitude [°E] | Altitude [m] | Land use | NEE time series for: Parameter estimation* & model evaluation | [+]tower |
|---|---|---|---|---|---|---|
| Merzenhausen (ME) | 50.9298 | 6.2970 | 93.3 | agriculture | 1 Dec. 2011 – 30 Nov. 2012* | Per.t |
| | | | | | 1 Dec. 2012 – 30 Nov. 2013 | |
| Selhausen (SE) | 50.8658 | 6.4474 | 103.0 | agriculture | 1 Dec. 2012 – 30 Nov. 2013 | Per.t |
| Niederzier (NZ) | 50.8795 | 6.4499 | 102.9 | agriculture | 5 Apr. 2013 – 10 Jul. 2013 | Rov.st. |
| Engelskirchen (EN) | 50.9115 | 6.3090 | 108.9 | agriculture | 18 Aug. 2012 – 24 Oct. 2012 | Rov.st. |

[+] Per.t.: Permanent EC tower; Rov.st.: roving station

**Table 2. Parameters applied for the perturbation of the meteorological input data, adapted from Han et al. (2014).**

| Variables [unit] | additive (A) or multiplicative (M) | Standard deviation | Cross correlation coefficients | | |
|---|---|---|---|---|---|
| | | | T | SW | LW |
| Air temperature (T) [K] | A | 1 | - | 0.4 | 0.4 |
| Incoming shortwave radiation (SW) [$Wm^{-2}$] | M | 0.3 | 0.4 | - | -0.5 |
| Incoming longwave radiation (LW) [$Wm^{-2}$] | A | 20 | 0.4 | -0.5 | - |
| Precipitation (P) [mm/s] | M | 0.5 | 0 | -0.8 | 0.5 |

**Table 3. Lower and upper bound of 95% confidence intervals of CLM4.5 parameter estimates estimated with DREAM$_{(zs)}$ (C3-crop) or adapted from Post et al. (2016)**

| | Year | $fl_{N_R}$ | $sla_{top}$ | $g_R$ | $r_b$ | $\psi_c$ | $Q_{10}$ | $mr_b$ | $b_s$ |
|---|---|---|---|---|---|---|---|---|---|
| CLM4.5 default values | | 0.14 | 0.030 | 0.30 | 2.00 | $-2.75 \times 10^5$ | 1.50 | $2.53 \times 10^{-6}$ | 9.0 |
| **RO** (C3-grass) | 11/'12 | 0.13, 0.15 | 0.010, 0.010 | 0.39, 0.40 | 1.01, 1.27 | $-3.79 \times 10^5$, $-1.65 \times 10^5$ | 1.39, 1.44 | $4.48 \times 10^{-6}$, $4.50 \times 10^{-6}$ | 6.1, 6.9 |
| CLM4.5 default values | | 0.18 | 0.030 | 0.30 | 3.00 | $-2.75 \times 10^5$ | 1.50 | $2.53 \times 10^{-6}$ | 9.0 |
| **ME** (C3-crop) | 11/'12 | 0.09, 0.10 | 0.010, 0.010 | 0.40, 0.40 | 1.00, 1.04 | -3.94E+05, -2.62E+05 | - | - | - |
| CLM4.5 default values | | 0.05 | 0.010 | 0.30 | 2.00 | $-2.55 \times 10^5$ | 1.50 | $2.53 \times 10^{-6}$ | 9.0 |
| **WÜ** (coniferous forest) | 11/'12 | 0.05, 0.07 | 0.005, 0.006 | 0.29, 0.40 | 0.75, 3.95 | $-3.91 \times 10^5$, $-2.07 \times 10^5$ | 2.50, 2.99 | $2.13 \times 10^{-6}$, $3.48 \times 10^{-6}$ | 5.0, 6.2 |
| CLM4.5 default values | | 0.05 | 0.010 | 0.30 | 2.00 | $-2.55 \times 10^5$ | 1.50 | $2.53 \times 10^{-6}$ | 9.0 |
| **FR-Fon** (deciduous forest) | 06/'07 | 0.12, 0.12 | 0.010, 0.010 | 0.39, 0.40 | 1.00, 1.17 | $-3.89 \times 10^5$, $-2.57 \times 10^5$ | 1.87, 1.97 | $3.47 \times 10^{-6}$, $3.50 \times 10^{-6}$ | 5.7, 6.0 |

**Table 4. Root mean square error RMSE$_m$ [µmol m$^{-2}$ s$^{-1}$], mean absolute difference for the mean diurnal NEE cycle MAD$_{diur}$ [µmol m$^{-2}$ s$^{-1}$], and relative difference of the NEE sum over the evaluation period RD$_{\sum NEE}$ [%] for the CLM ensemble with estimated parameters (Ens$_P$) in comparison to the reference run (Ref) with default parameters. Results are given for ME (Merzenhausen), SE (Selhausen), NZ (Niederzier) and EN (Engelskirchen). Here, *n* is the amount of non-gap filled half-hourly measurement data that was available to calculate these evaluation indices.**

| grid cells | *n* | MAD$_{diur}$ [µmol m$^{-2}$ s$^{-1}$] | | RMSE [µmol m$^{-2}$ s$^{-1}$] | | RD$_{\sum NEE}$ [%] | |
|---|---|---|---|---|---|---|---|
| | | Ens$_P$ | Ref | Ens$_P$ | Ref | Ens$_P$ | Ref |
| **ME** | *10157* | 1.5 | 2.30 | 5.95 | 6.74 | 11 | 79 |
| **SE** | *9597* | 3.22 | 3.45 | 8.20 | 9.41 | 81 | 94 |
| **NZ** | *2772* | 2.70 | 3.61 | 10.71 | 9.37 | 33 | 99 |
| **EN** | *2272* | 4.83 | 4.53 | 8.83 | 8.93 | 87 | 82 |

**Table 5. Mean leaf area indices (LAI$_{PFT}$) for the RapidEye data (Obs), the CLM ensembles with estimated parameters (Ens$_P$) and the CLM reference run (Ref) with default parameters, as well as the mean absolute differences (MAD$_{LAI}$) and root mean square error (RMSE$_{LAI}$). $N$ is the number of grid cells in the catchment covered by more than 80% by one specific PFT.**

| PFT | | | LAI$_{PFT}$ | | | RMSE$_{LAI}$ | | MAD$_{LAI}$ | |
|---|---|---|---|---|---|---|---|---|---|
| | | $N$ | Obs | Ens$_P$ | Ref | Ens$_P$ | Ref | Ens$_P$ | Ref |
| C3-grass | w | 1667 | 2.7±1.6 | 1.0±0.8 | 2.6±2.2 | 2.6 | 3.0 | 2.0 | 2.5 |
| C3-grass | s | 1271 | 3.0±1.4 | 1.8±0.6 | 5.5±1.4 | 1.9 | 3.2 | 1.5 | 2.6 |
| C3-crop | w | 4392 | 2.3±1.3 | 0.7±0.5 | 2.4±1.7 | 2.1 | 1.9 | 1.7 | 1.6 |
| C3-crop | s | 2887 | 2.1±0.9 | 1.5±0.5 | 5.5±0.8 | 1.0 | 3.6 | 0.8 | 3.5 |

5    w: winter half year (Nov.-Apr.), s: summer half year (May-Oct.), ± Standard deviation, Ens$_P$: CLM ensemble with estimated parameters

**Figures**

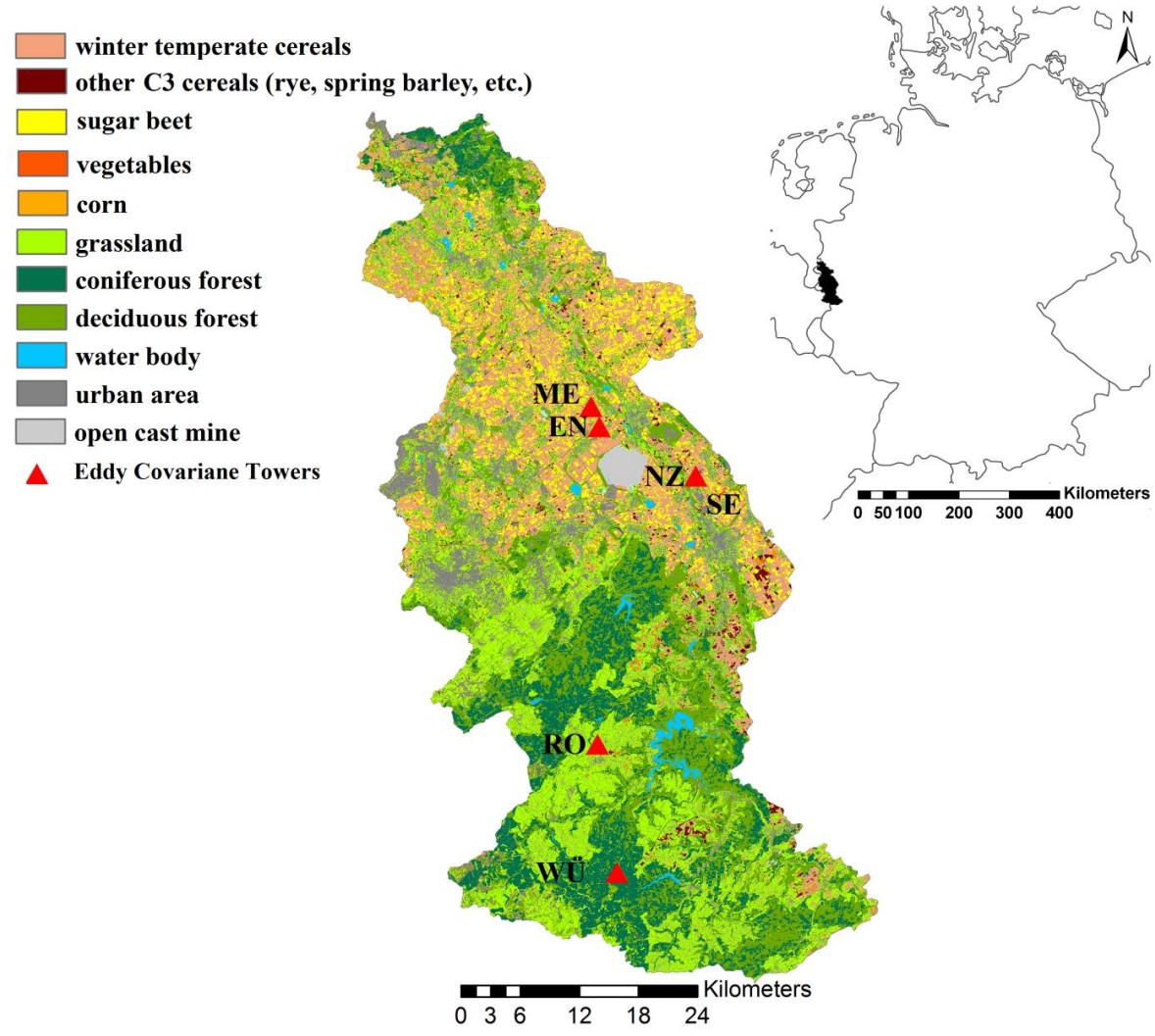

Figure 1. Land cover (Waldhoff, 2010) and eddy covariance tower sites in the Rur catchment.

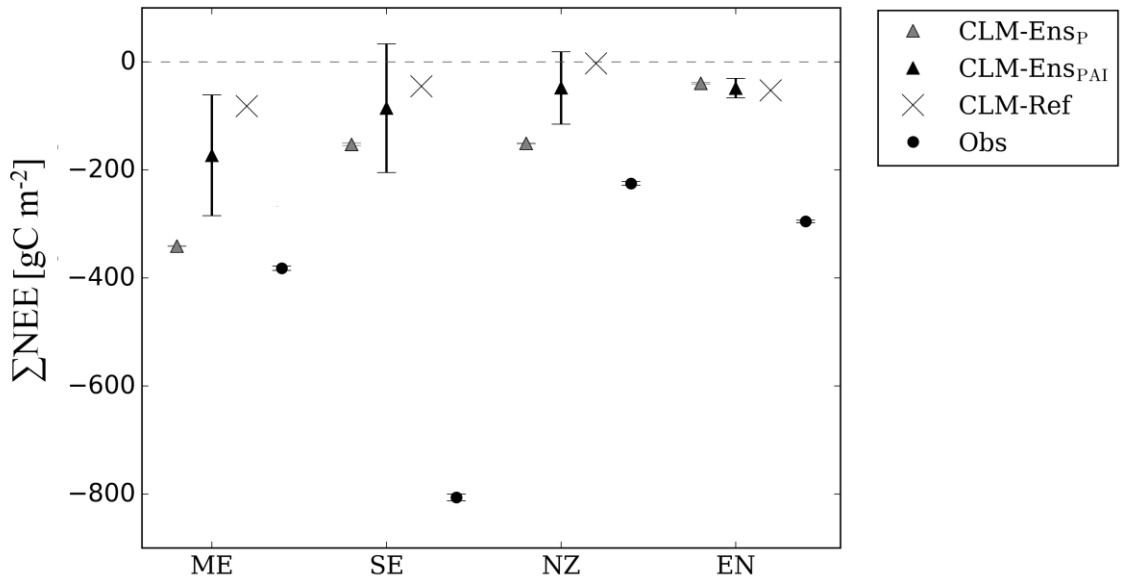

**Figure 2. NEE sum ($\sum$NEE) determined for the evaluation periods (see Table 1) including all half-hourly time steps where eddy covariance data (Obs) were available, for the CLM ensembles $Ens_P$ with estimated parameters, and $Ens_{PAI}$ with additional perturbed atmospheric forcings and perturbed initial states (including standard deviations of the 60 ensemble members), in comparison to a reference run with default parameters (CLM-Ref). In all CLM cases time series were subsetted according to available Obs.**

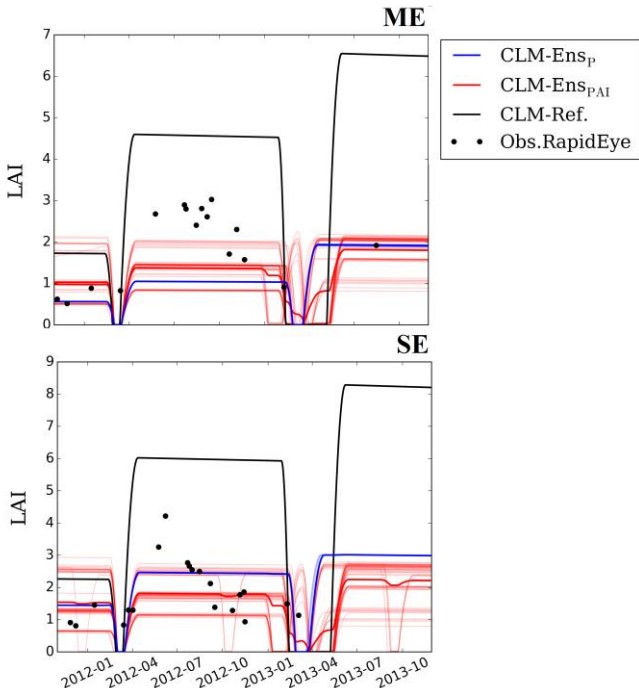

**Figure 3. Daily leaf area indices (LAI) for the evaluation period 1 Dec 2011- 30 Nov 2013 for grid cells where the sites Merzenhausen (ME), and Selhausen (SE) are located. Results are shown for the 60 ensemble members of the CLM cases Ens_PertP with estimated parameters, and Ens_PAI with additional perturbed atmospheric forcings and perturbed initial states, in comparison to a reference run with default parameters (CLM-Ref) and RapidEye data (Obs.RapidEye) (Bold lines: ensemble mean).**

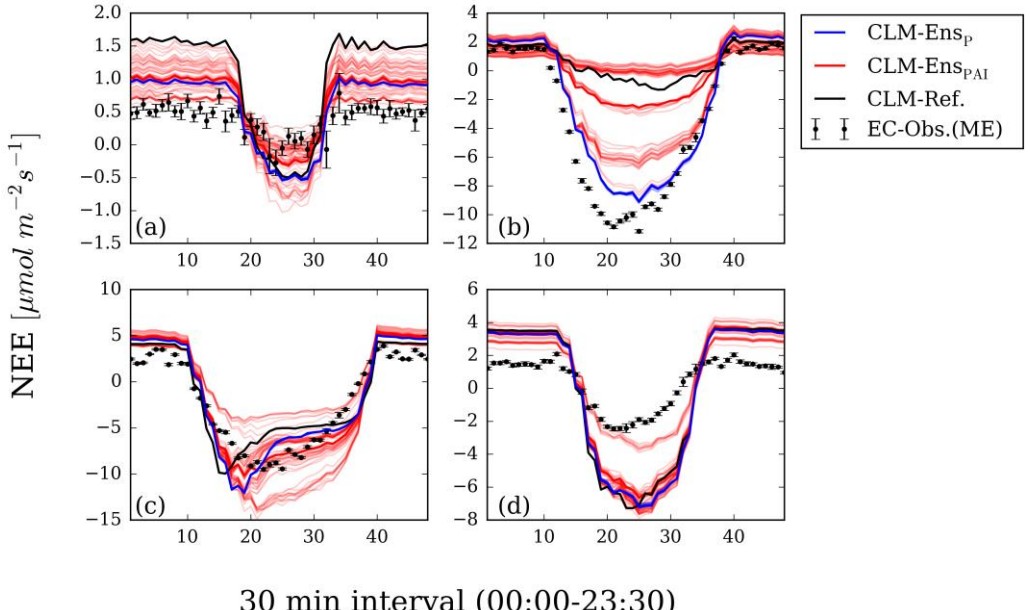

**Figure 4.** Mean diurnal course of half-hourly NEE for winter '12/'13 (a), spring 2013 (b), summer 2013 (c) and autumn 2013 (d) for the Merzenhausen site (ME). Results are shown for the 60 ensemble members of the CLM cases Ens$_P$ with estimated parameters, and Ens$_{PAI}$ with additional perturbed atmospheric forcings and perturbed initial states, in comparison to a reference run with default parameters (CLM-Ref) and EC data (EC-Obs.) (Bold lines: ensemble mean).

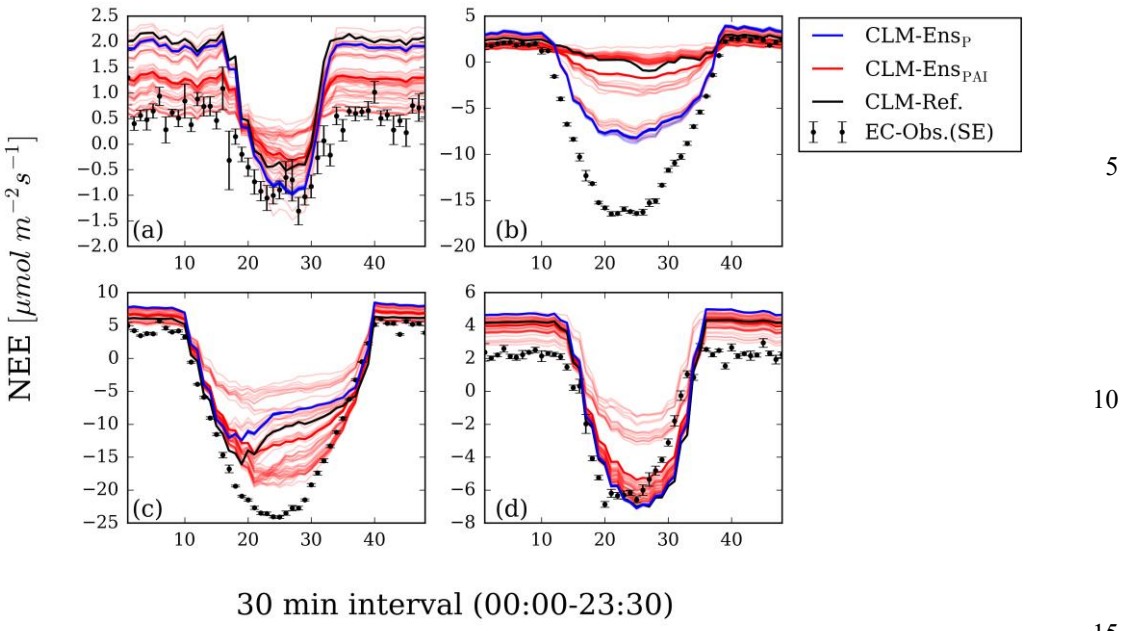

**Figure 5.** Like Figure 4, for the Selhausen site (SE).

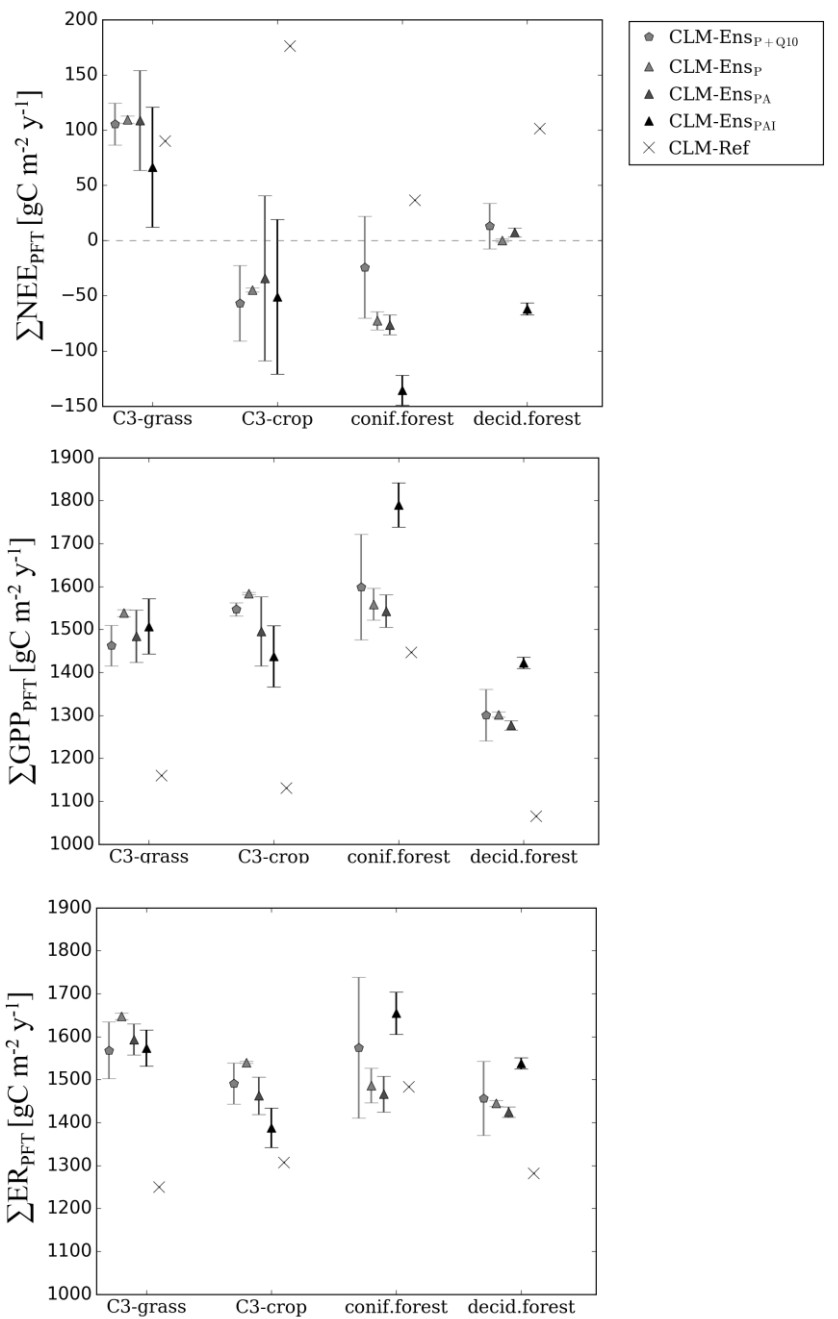

**Figure 6. Ensemble mean and standard deviation of carbon fluxes simulated by four different CLM ensembles: (i) with estimated parameters (Ens$_P$) and (ii) additional perturbation of Q$_{10}$ (Ens$_{P+Q10}$) or atmospheric forcings (Ens$_P$) or atmospheric forcings and initial states (Ens$_{PAI}$), in comparison to the reference run with default parameters (Ref). Plotted are the PFT-specific, catchment average sums of NEE, GPP and ER.**

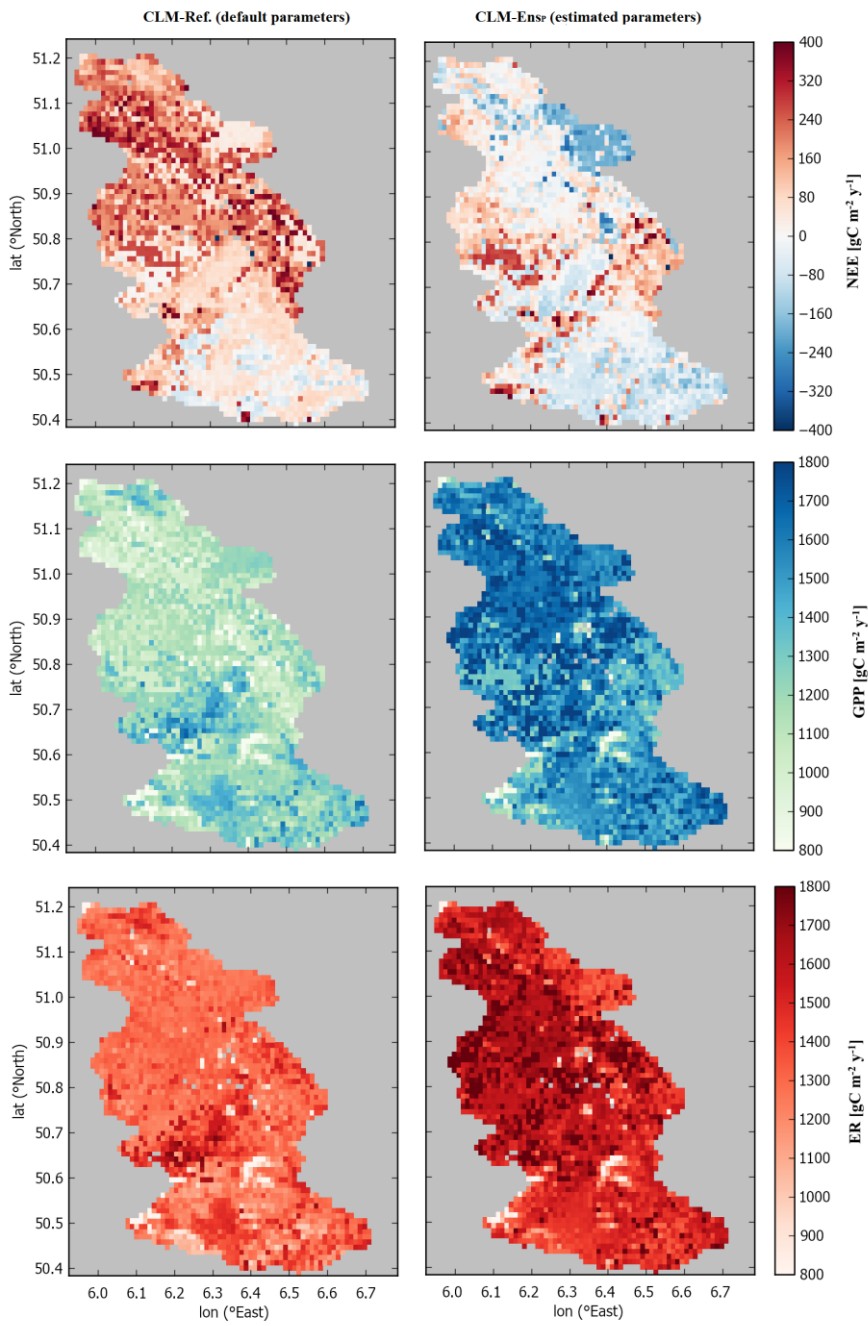

**Figure 7. Annual sum of net ecosystem exchange (NEE), gross primary production (GPP) and ecosystem respiration (ER) determined with CLM4.5-BGC for the Rur catchment (Dec. 2012-Nov. 2013) with default parameters (CLM-Ref.) and with estimated parameters (CLM-Ens).**