# Peer review of "Evaluation and uncertainty analysis of regional scale CLM4.5 net carbon flux estimates"

_Biogeosciences, 2016_

## Referee Comment (RC1) · Anonymous Referee #1 · 13 Feb 2017

In this manuscript, the authors apply the well-known CLM land surface model to simulate ecosystem carbon fluxes at the catchment-scale. The main novelty of this article is the uncertainty analysis that is undertaken, using state-of-the-art MCMC procedure to evaluate model parameter uncertainty, but also introducing uncertainties in environmental drivers and initial conditions. For each step, the uncertainty in the predictions is matched with a default simulations that uses "out-of-the-box" parameter values for the various PFTs that can be found over the Rur catchment.

This article is relevant to the readership of Biogeosciences as it aims to bridge a gap that exists between eddy-covariance based approaches, and large-scale modelling contributions. However, I am concerned by the shortness of the discussion, and the lack of references to other groups' work therein. It makes it hard to put this piece of work within the current context. As the authors correctly mention, regional-scale un-

certainty analyses of land surface models are rare, but the various points they raise in the results sections have already been addressed in various studies. For example, the sensitivity of net carbon fluxes to Q10 value, or the various environmental response functions that affect turnover rates, has been addressed in detail by Davidson and Janssens (2006), Todd-Brown et al. (2013, 2014), Exbrayat et al. (2014a,b). Furthermore, the transferrability of PFT parameters has been studied by Kuppel et al. (2014).

Hereafter are some more specific points:

p3 l25: see also Xia et al. (2012) / Exbrayat et al. (2014a);

p3 l27: please consider adding a reference to the Global Carbon Budget / TRENDY intercomparison project here

p5 l2: please indicate time series length here

p5 l10-12: although this has been reported in a previous paper, can you please develop a bit more on how parameters were optimised for these sites?

p6 l13: can you indicate the respective fraction of each PFT here instead of l25

p7 l10: how representative of mean climate were the three years used for spin-up?

p7 l11: what is an "exit" spin-up? does that simply mean that the spin-up was 1203 years repeated meteorological drivers for 2008-2010

p7 l24: what value of the convergence diagnostic did you use?

p8 l9: please add reference for Fontainebleau's site

p8 l26: compared

p10 l1: please add full name and references for the two sites

p11 l19: are 15 years enough for this perturbed spin-up?

p12 l8: I am not clear what "deterministic initial states" mean here

p13 l1: why are results for ENSp outside the range from ENSpai? from the experiment description it feels that ENSpai's uncertainty should envelop that of ENSp (and ENSpa)

p13 l21: there is a problem with figure numbering.

p13 l23: daytime NEE is not GPP, please replace/revise in the rest of the paragraph

p 15 l17: the discussion is very short in regards of the amount of results that are described.

p16 l1: a new alternative to the PFT approach is to rely on more continuously distributed trait maps (eg Castanho et al., 2013, Reichstein et al., 2014)

Figures 3,4 and 5: please correct the legend

Castanho, A. D. A., Coe, M. T., Costa, M. H., Malhi, Y., Galbraith, D. and Quesada, C. A.: Improving simulated Amazon forest biomass and productivity by including spatial variation in biophysical parameters, Biogeosciences, 10, 2255–2272, doi:10.5194/bg-10-2255-2013, 2013.

Davidson, E. A. and Janssens, I. A.: Temperature sensitivity of soil carbon decomposition and feedbacks to climate change., Nature, 440(7081), 165–73, doi:10.1038/nature04514, 2006.

Exbrayat, J.-F., Pitman, A. J. and Abramowitz, G.: Response of microbial decomposition to spin-up explains CMIP5 soil carbon range until 2100, Geosci. Model Dev., 7(6), 2683–2692, doi:10.5194/gmd-7-2683-2014, 2014a.

Exbrayat, J.-F., Pitman, A. J. and Abramowitz, G.: Disentangling residence time and temperature sensitivity of microbial decomposition in a global soil carbon model, Biogeosciences, 11(23), doi:10.5194/bg-11-6999-2014, 2014b.

Kuppel, S., Peylin, P., Maignan, F., Chevallier, F., Kiely, G., Montagnani, L. and Cescatti, A.: Model–data fusion across ecosystems: from multisite optimizations to global simulations, Geosci. Model Dev., 7(6), 2581–2597, doi:10.5194/gmd-7-2581-2014, 2014.

Reichstein, M., Bahn, M., Mahecha, M. G., Kattge, J. and Baldocchi, D. D.: Linking plant and ecosystem functional biogeography, PNAS, 111(38), 13697-13702, doiL10.1073/pnas.1216065111, 2014.

Todd-Brown, K. E. O., Randerson, J. T., Post, W. M., Hoffman, F. M., Tarnocai, C., Schuur, E. a. G. and Allison, S. D.: Causes of variation in soil carbon simulations from CMIP5 Earth system models and comparison with observations, Biogeosciences, 10(3), 1717–1736, doi:10.5194/bg-10-1717-2013, 2013.

Todd-Brown, K. E. O., Randerson, J. T., Hopkins, F., Arora, V., Hajima, T., Jones, C., Shevliakova, E., Tjiputra, J., Volodin, E., Wu, T., Zhang, Q. and Allison, S. D.: Changes in soil organic carbon storage predicted by Earth system models during the 21st century, Biogeosciences, 11(8), 2341–2356, doi:10.5194/bg-11-2341-2014, 2014.

Xia, J. Y., Luo, Y. Q., Wang, Y.-P., Weng, E. S. and Hararuk, O.: A semi-analytical solution to accelerate spin-up of a coupled carbon and nitrogen land model to steady state, Geosci. Model Dev., 5(5), 1259–1271, doi:10.5194/gmd-5-1259-2012, 2012.

---

## Referee Comment (RC2) · S. Zaehle (Referee) · 24 Apr 2017

Dear authors,

first - my apologies in the delay of the review process. After failing to receive sufficient reviews, I now provide a brief assessment myself.

I find this a very interesting and robust study, which unfortunately still needs a bit of fine tuning, in particular in the presentation. By reading the abstract I was under the impression that this study was "merely" an application of a data assimilation method result to larger scales, whereas in truth the study expands to uncertainty in met-forcing and initial conditions, which is typically ignored in large-scale applications. I think this achievement deserves a bit more presentation in abstract, results and also discussion.

In the introduction, the text could be streamlined to focus on the current study and avoid side-lines as the use of atmospheric inversions or the future predicted climate of the Rur valley. I've been missing an introduction section that addresses in some more details the motivation for not only using DA to obtain model parameters, but at the same time use of perturbed boundary conditions (meteorological and soil), and a discussion of this in Section 4. I think this is really a novel contribution, which deserves somewhat more place.

Methodologically, I am unsure whether the assimilation procedure included the spin-up period, or not? In the latter case, one should caution the interpretation of the effect of the assimilation on the cumulative NEE?

The discussion is simply too cursory and needs to better take account of existing literature and explaining the added insights gained from this study. This could be partially achieved by taking considerations from the conclusions and expanding these ideas in the light of the existing literature (or potential applications for regional modelling).

Minor comments:

The abstract is quite long, consider shortening.

P1 L19: give average error reduction in $\mu$mol CO2 / m2 / s ?

P1 L21: a) is this in agreement with the observations (the fact that the NEE goes positive); b) add "simulated" before regional carbon balance estimates

P1 L22: here and elsewhere it would be important to understand whether you have brought the model into equilibrium in terms of the carbon cycle with the parameter set used, or whether you relied on a different way to initialise carbon stocks

P1 L26: It would be helpful to add in brackets the difference in annual integrated NEE in $\mu$mol / m2 / year

P1 L29: If the uncertainty was indeed reduced, then this would need to be demonstrated by comparing prior and posterior distributions of the regional NEE. I guess you rather would like to state that the accuracy of the projection has increased because the model better fits site-level observations?

P2 L11: define what "conventional interpolation methods" are

P2 L12ff: all correct, but I don't think this paragraph is necessary for the sake of this paper.

P3 L2: please define more precisely what "error" is here. There is no principle error in averaging the input data, and then obtaining model output from this. The question is whether the aggregation method is sufficiently representing the average regional flux, is of course relevant.

P3 L32: I disagree that the uncertainty of carbon fluxes has been overlooked so far. I agree that the uncertainty hasn't been sufficiently quantified, and/or reduced.

P4L1: well, if you counted conference papers, it actually has. I don't think the question of "who was first" is really relevant, and would focus more on the value and design of the regional set-up

P4 L6: define what a "validated parameter" is. I assume that this is a parameter derived from data-assimilation? It is unclear to me why new estimates of parameters have been obtained for one PFT. Is this because you extended your PFT set to a new PFT (from 2016), or because the DA method of 2016 did not yield good parameters. In the latter case, would it not have been appropriate to recalibrate the model for all sites?

P6 L24: add "average" before percentage PFT cover.

P7 L11: Please briefly explain, why it was necessary to add a second spin-up

P7 L22: The statement of robustness of the method either requires a proof or (more likely) a reference

P11 L 25: I make this note here, although this probably needs mentioning later in that

by not manipulating the "stable" nitrogen pool you basically conserve the amount of N available from net N mineralisation across the ensemble. That's fine (in particular, because the system are fertilised), but probably implies that you underestimate the true uncertainty in the net N availability at the sites.

P15 L7 (and elsewhere): please refer to other parts of the manuscript as sections, not chapters.

P15L11: Please be more precise, this is not really a general finding, it is specific to Q10.

Please have the revised manuscript cross-checked by a native speaker, and remove unnecessary words (such as "however" in the middle of a sentence, e.g. P15 L8).

P15: This discussion section is a bluntly speaking a bit too skinny. More could be made for instance of the extremely relevant and very nicely demonstrated point that minor modifications in input meteorology lead to markable differences in modelled NEE. However, generally, a critical assessment of the novelty of the study approach and finding compared to the existing literature is missing.

Table 3: Please arrange the table such that it is immediately clear which CLM4.5 default simulation belongs to which site

Table 4: please briefly explain the abbreviations used to denote "grid cells", and how the "n" were calculated (or what they are).

Table 5: define "n"

Figure 2: define "evaluation period". It would also be preferable to somewhat make a clearer distinction between different model ensembles and the observations

---

## Author Comment (AC1) · 20 Jun 2017

**Author's response**

We highly appreciate the constructive comments and literature advice to further improve the paper. We considered all suggestions as prescribed below. The different colors indicate: (grey:) editors comment and (black:) author's response.

As the authors correctly mention, regional-scale uncertainty analyses of land surface models are rare, but the various points they raise in the results sections have already been addressed in various studies. For example, the sensitivity of net carbon fluxes to Q10 value, or the various environmental response functions that affect turnover rates, has been addressed in detail by Davidson and Janssens (2006), Todd-Brown et al. (2013, 2014), Exbrayat et al. (2014a,b). Furthermore, the transferrability of PFT parameters has been studied by Kuppel et al. (2014).

It is correct that other studies like these investigated aspects that we investigated too in this paper. However, these studies did not estimate the uncertainty of modelled region scale NEE as we did herein, taking into account the uncertainty of forcings data, initial states and parameter values. We will address this comment and point out the differences including references in the revised version of the manuscript.

p3 l25: see also Xia et al. (2012) / Exbrayat et al. (2014a);

We will add the suggested literature references.

p3 l27: please consider adding a reference to the Global Carbon Budget / TRENDY intercomparison project here

We will add the suggested reference.

p5 l2: please indicate time series length here

Time series lengths are summarized in Table 1. We will indicate this in this sentence: "Eddy covariance (EC) data in the Rur catchment were measured four C3-crop sites for the evaluation periods summarized in Table 1."

p5 l10-12: although this has been reported in a previous paper, can you please develop a bit more on how parameters were optimized for these sites?

In section 2.3.1 we explained how parameters were optimized and refer to the previous paper for further details. As the focus of this paper is not on parameter estimation but on model uncertainty (which we will now better highlight in the abstract), we would not extent the outline of the parameter estimation methods. If there is some particular information you are missing, we can add it.

p6 l13: can you indicate the respective fraction of each PFT here instead of l25

The sentence "The percentage PFT coverage of the vegetated land in Rur catchment was ~34% C3-crops, ~32% grassland, ~17% broadleaf deciduous forest and ~14% coniferous forest." was moved upward as suggested.

p7 l10: how representative of mean climate were the three years used for spin-up?

According to http://www.dwd.de/DE/klimaumwelt/klimaatlas/klimaatlas_node.html , the annual average temperature in the years 2008 and 2009 was ~ 0.5-1.0 °C higher the long term average (1961-1990), the mean annual temperate in 2010 was ~ 0.5-1.0 °C lower. Mean precipitation amounts and freezing days where representative for the long term average. Before conducting this study, we extensively studied the effect of the forcing data used for the model spin-up. For example, we tested using a longer time series (1998-2004) of global climate data. However, we found that using more recent, regional forcing data during the spin-up resulted in carbon- and energy fluxes in better agreement with the observations. This will be clarified in the modified version of the manuscript.

p7 l11: what is an "exit" spin-up? does that simply mean that the spin-up was 1203 years repeated meteorological drivers for 2008-2010

Principally, yes. We reformulated this sentence. "Exit spin-up" is a technical term for CLM. Before conducting the actual model runs with the restart files from the model spin-up, an intermediate run (the exit spin-up") is

necessary with specific parameter settings. This is e.g. because CLM uses an accelerated model spin-up mode. We will clarify in the paper: "The model states obtained after the 1200-year spin-up were then used as input for a second three years "exit spin-up" also using the meteorological data for the years 2008-2010. The "exit spin-up" in CLM is necessary for technical reasons and switches the CLM settings from the (accelerated) spin-up mode to the "normal" mode in terms of the calculated carbon-nitrogen cycling."

p7 l24: what value of the convergence diagnostic did you use?

We used a threshold of 1.2 to declare convergence as suggested e.g. in Vrugt (2015).

p8 l9: please add reference for Fontainebleau's site

We can add a reference. Is there a particular reference you can suggest?

p8 l26: compared

This will be corrected

p10 l1: please add full name and references for the two sites. Are there particular references you can suggest?

We will add it.

p11 l19: are 15 years enough for this perturbed spin-up?

We think so. We also tested longer perturbed spin-up periods up to 30 years and found there were no significant differences of the ensemble spread. We found that after about 8 years, the ensemble spread reaches some kind of maximum and than does not increase considerably further.

p12 l8: I am not clear what "deterministic initial states" mean here

We added "(non-perturbed)" in the sentence (p. ll.): CLM-EnsP: ensemble model runs for 1 Dec 2012 – 30 Nov 2013 with deterministic (non-perturbed) initial states and non-perturbed input data from COSMO-DE (Sect.2.2)." to clarify that we mean the default, non perturbed initial states obtained with the restart files from the 1203 years non perturbed spin-up.

p13 l1: why are results for ENSp outside the range from ENSpai? from the experiment description it feels that ENSpai's uncertainty should envelop that of ENSp (and ENSpa)

In case of $ENS_{PAI}$ perturbed atmospheric forcing AND perturbed initial states were applied in contrast to $ENS_P$. For $ENS_P$, the non-perturbed forcing and initial state data sets were used. The large discrepancy between $ENS_{PAI}$ and $ENS_P$ is one of the major findings of this paper and highlights the overly strong sensitivity of the modeled carbon pools and fluxes to the climate forcings.

We will extend the discussion on that.

p13 l21: there is a problem with figure numbering.

Which problem? We checked the sentence and it seems correct/ as intented.

p13 l23: daytime NEE is not GPP, please replace/revise in the rest of the paragraph.

This of course is correct. We revised the paragraph (p. , ll. ): "The delay of the plant emergence indicated by these LAI courses is related to the strong underestimation of daytime NEE in spring 2013 (Figure 3, Figure 4). This underestimation of NEE can mainly be attributed to an underestimation of GPP, which is too low in spring because the simulated plant onset was about 2 weeks later than observed. For CLM-Ref and for most of the CLM-EnsPAI ensemble members, leaf onset started in May. For those model realizations, the underestimation of daytime NEE in spring was highest. In contrast, for CLM-EnsP and a small proportion of CLM-EnsPAI, leaf onset already started in March. For those cases, the underestimation of daytime NEE in spring was notably lower. This elucidates the close link of modelled NEE and LAI and highlights that errors in the timing of leaf onset can lead to substantial errors in simulated NEE. The evaluation of simulated LAI showed that modelled

and observed LAI are closer for simulations with estimated CLM parameters than for simulations with default parameters and that in case of C3-crop, simulated leaf onset was better represented."

p 15 l17: the discussion is very short in regards of the amount of results that are described.

We agree and will extend the discussion, including the references you suggested.

Figures 3, 4 and 5: please correct the legend

The legend has been corrected.

---

## Author Comment (AC2) · 20 Jun 2017

**Author's response**

**We highly appreciate the constructive comments and literature advice to further improve the paper. We considered all suggestions as prescribed below. The different colors indicate: (grey:) editors comment and (black:) author's response.**

I find this a very interesting and robust study, which unfortunately still needs a bit of fine tuning, in particular in the presentation. By reading the abstract I was under the impression that this study was "merely" an application of a data assimilation method result to larger scales, whereas in truth the study expands to uncertainty in metforcing and initial conditions, which is typically ignored in large-scale applications. I think this achievement deserves a bit more presentation in abstract, results and also discussion.

**Thanks. We will revise the abstract, results and discussion accordingly.**

In the introduction, the text could be streamlined to focus on the current study and avoid side-lines as the use of atmospheric inversions or the future predicted climate of the Rur valley.

**The introduction has been revised as suggested.**

I've been missing an introduction section that addresses in some more details the motivation for not only using DA to obtain model parameters, but at the same time use of perturbed boundary conditions (meteorological and soil), and a discussion of this in Section 4. I think this is really a novel contribution, which deserves somewhat more place.

**We agree and will extend the introduction and discussion sections to address this.**

Methodologically, I am unsure whether the assimilation procedure included the spin-up period, or not? In the latter case, one should caution the interpretation of the effect of the assimilation on the cumulative NEE?

**This point is not clear to us. Can the reviewer please clarify this?**

The discussion is simply too cursory and needs to better take account of existing literature and explaining the added insights gained from this study. This could be partially achieved by taking considerations from the conclusions and expanding these ideas in the light of the existing literature (or potential applications for regional modelling).

**We agree that the discussion was too short and will extend/extended it as suggested.**

Minor comments:

The abstract is quite long, consider shortening.

**The abstract is revised now and somewhat shorter than before.**

P1 L19: give average error reduction in mol CO2 / m2 / s?

**We did not include absolute error reductions here, because we used non gap filled data to calculate the NEE sum. The absolute values here may be misleading.**

p1 L21: a) is this in agreement with the observations (the fact that the NEE goes positive); b) add "simulated" before regional carbon balance estimates

**a) We do not say that NEE itself but $\sigma_{\Sigma NEE}$ (the model uncertainty of the NEE sum) did increase, b) "simulated" is added as suggested.**

P1 L22: here and elsewhere it would be important to understand whether you have brought the model into equilibrium in terms of the carbon cycle with the parameter set used, or whether you relied on a different way to initialise carbon stocks

As shown e.g. in Post et al. (2016), parameters strongly depend on the set of initial conditions. A main conclusion of this paper was that strictly speaking, estimated CLM parameters cannot necessarily be considered valid when combined with initial states very different from the states that were used during parameter estimation and model evaluation. The strong correlation of initial states like carbon-nitrogen pools and parameters is a central problem in the current usage of complex land surface models. Unfortunately, due to the high computational demand, it is not possible include the spin-up in the parameter estimation procedure for land surface models like CLM.

P1 L26: It would be helpful to add in brackets the difference in annual integrated NEE in mol / m2 / year

**We will add this as suggested.**

P1 L29: If the uncertainty was indeed reduced, then this would need to be demonstrated by comparing prior and posterior distributions of the regional NEE. I guess you rather would like to state that the accuracy of the projection has increased because the model better fits site-level observations?

Since we did not present a reduction of model uncertainty by comparing the prior and posterior uncertainty, we agree this sentence should be reformulated. Indeed, the uncertainty of modeled NEE that can be attributed to the parameter uncertainty was reduced. Figure 6 highlights how the model uncertainty increases if only one key parameter (here  $Q_{10}$ ) is perturbed. We are going to rephrase the conclusions here.

P2 L11: define what "conventional interpolation methods" are

We meant simple spatial interpolation methods and can modify or delete this sentence.

P2 L12ff: all correct, but I don't think this paragraph is necessary for the sake of this paper.

**We intended to highlight the importance of regional scale and high resolution LSM modeling approaches to examine and predict carbon flux dynamics and uncertainties, as it was done study, but can delete this paragraph.**

P3 L2: please define more precisely what "error" is here. There is no principle error in averaging the input data, and then obtaining model output from this. The question is whether the aggregation method is sufficiently representing the average regional flux, is of course relevant.

**We will reformulate: "The model-data discrepancy as well as the reliability of model input data"**

P3 L32: I disagree that the uncertainty of carbon fluxes has been overlooked so far. I agree that the uncertainty hasn't been sufficiently quantified, and/or reduced.

**This was mainly referring to regional or larger scale modeling studies. We will reformulate the sentence.**

P4L1: well, if you counted conference papers, it actually has. I don't think the question of "who was first" is really relevant, and would focus more on the value and design of the regional set-up

**We agree that the question of "who is first" is not relevant but just intended to highlight the significance of our study in this context. We will delete this sentence.**

P4 L6: define what a "validated parameter" is. I assume that this is a parameter derived from data-assimilation? It is unclear to me why new estimates of parameters have been obtained for one PFT. Is this because you extended your PFT set to a new PFT (from 2016), or because the DA method of 2016 did not yield good parameters. In the latter case, would it not have been appropriate to recalibrate the model for all sites?

"Validated parameter estimates" refers to the parameter estimates which have been successfully estimated and evaluated in the previous study, i.e. yielded NEE model outputs closer to the measured values than the default CLM parameter values. Correct, for one PFT (C3-crop) the estimated parameter values from the previous study could not be successfully evaluated, i.e. did not clearly reduce the model-data misfit. Therefore, we estimated new parameter values for C3-crop in this study.

P6 L24: add "average" before percentage PFT cover.

It is added now.

**P7 L11: Please briefly explain, why it was necessary to add a second spin-up**

This is the default procedure suggested in the CLM user manual and is mainly necessary due to technical reasons. We will clarify in the paper:: "The model states obtained after the 1200-year spin-up were then used as input for a second three years "exit spin-up" also using the meteorological data for the years 2008-2010. The "exit spin-up" in CLM is necessary for technical reasons and switches the CLM settings from the (accelerated) spin-up mode to the "normal" mode in terms of the calculated carbon-nitrogen cycling."

P7 L22: The statement of robustness of the method either requires a proof or (more likely) a reference

**We added "(Vrugt, 2015)".**

P11 L 25: I make this note here, although this probably needs mentioning later in that by not manipulating the "stable" nitrogen pool you basically conserve the amount of N available from net N mineralisation across the ensemble. That's fine (in particular, because the system are fertilised), but probably implies that you underestimate the true uncertainty in the net N availability at the sites.

**We agree and can add a sentence in this respect.**

P15 L7 (and elsewhere): please refer to other parts of the manuscript as sections, not chapters.

**We will modify this.**

P15L11: Please be more precise, this is not really a general finding, it is specific to Q10. Please have the revised manuscript cross-checked by a native speaker, and remove unnecessary words (such as "however" in the middle of a sentence, e.g. P15 L8).

**We will correct this and carefully recheck the manuscript for unnecessary words and grammatical errors.**

P15: This discussion section is a bluntly speaking a bit too skinny. More could be made for instance of the extremely relevant and very nicely demonstrated point that minor modifications in input meteorology lead to markable differences in modelled NEE.

**We agree that the discussion is too short and we will extend it.**

However, generally, a critical assessment of the novelty of the study approach and finding compared to the existing literature is missing.

It was difficult to compare our findings with the existing literature given the very few studies which evaluated LSM uncertainty in a similar way as was done here for carbon fluxes. However, we will recheck the literature again and try to extend the comparison with existing literature in the discussion section of the paper.

Table 3: Please arrange the table such that it is immediately clear which CLM4.5 default simulation belongs to which site

**We will re-organize the table.**

Table 4: please briefly explain the abbreviations used to denote "grid cells", and how the "n" were calculated (or what they are).

**In Table 4 "n" is the amount of non-gap filled half-hourly measurement data that was available to calculate these evaluation indices. We will clarify this.**

Table 5: define "n"

**Here, n in the number of grid cells in the catchment covered by more than 80% by one specific PFT. We will clarify this in the paper.**

Figure 2: define "evaluation period". It would also be preferable to somewhat make a clearer distinction between different model ensembles and the observations

The dates of the evaluation periods are summarized in Table 1.

---

## Author Response (AR1)

We highly appreciate the constructive comments and literature advice to further improve the paper. We considered all suggestions as prescribed below. The different colors indicate: (grey) editors comment and (red) author's response.

**REVIEWER 1**

As the authors correctly mention, regional-scale uncertainty analyses of land surface models are rare, but the various points they raise in the results sections have already been addressed in various studies. For example, the sensitivity of net carbon fluxes to Q10 value, or the various environmental response functions that affect turnover rates, has been addressed in detail by Davidson and Janssens (2006), Todd-Brown et al. (2013, 2014), Exbrayat et al. (2014a,b). Furthermore, the transferrability of PFT parameters has been studied by Kuppel et al. (2014).

It is correct that other studies like these investigated aspects that we investigated too in this paper. Four of the papers mentioned by the reviewer were also already cited in our work. However, these studies did not estimate the uncertainty of modelled region scale NEE as we did herein, taking into account the uncertainty of forcing data, initial states and parameter values. Therefore, the introduction has largely been reformulated to better point out the main contributions of this paper. In particular, the following text block which has been added is important (from P3,L16):

"Some studies estimate the uncertainty of terrestrial carbon flux predictions based on an ensemble of many different LSMs (e.g. Fisher et al., 2014; Huntzinger et al., 2012; Piao et al., 2013; Zhao et al., 2016). These studies focus on differences between models and therefore model structural deficits. Those studies highlight that (i) carbon flux predictions are generally highly uncertain, which also contributes to the uncertainty in climate change predictions, (ii) interactions of the different processes and drivers is not understood satisfactorily, and (iii) models require structural improvement to produce more consistent predictions. In order to improve LSM model structure and thus model-data and inter-model consistency, a more comprehensive understanding of model functionality and the contribution and link of the different model error sources is required. However, as highlighted by Xiao et al. (2014), the uncertainty of carbon fluxes obtained by ecosystem and land surface models has not been analyzed and quantified enough, particularly in regional scale studies.

Whereas the uncertainty of land surface model parameters has been subject to intensive investigations (e.g., Ren et al., 2013; Xiao et al., 2014), and several works are dedicated to reducing this uncertainty, for example by parameter estimation with data assimilation methods (e.g., Safta et al., 2015), the other sources of uncertainty are less studied. It was already concluded in earlier work that parameter uncertainty alone cannot explain observed deviations between measured and simulated NEE (e.g., Pridhoko et al., 2008; Wang et al., 2011), and the remaining deviations are often attributed to model structural errors. However, also uncertainty in atmospheric forcings and initial conditions could contribute to unexplained deviations between simulated and measured NEE. From global scale studies it is known that NEE is sensitive to the climate scenario. For example, in a study the LSM LPJ-GUESS was forced with output from 18 different coupled atmosphere-ocean global circulation models or earth system models, and showed very different NEE-responses depending on the climate scenario. Ten out of 18 models project that the land will become a carbon source in the 21$^{st}$ century, while the other eight models indicate that the land will act as a carbon sink (Ahlstrom et al., 2012). At the plot, catchment or regional scale and for shorter time periods in the recent past, very few studies analyzed the impact of uncertainty in meteorological forcings on NEE. Studies which analyzed interannual variability in NEE detected the important role of temperature and precipitation as drivers of this variability (e.g., Keppel-Aleks et al., 2014). Gu et al. (2016) found that for the Missouri Ozark AmeriFlux forest site, the interannual NEE-variability is smaller in simulations by CLM than in the data. On the other hand, some field studies for pairs of monitoring sites (Kwon et al., 2006) or experimentation sites (with temperature increase and precipitation increase or decrease) (Xu et al., 2016) found a limited impact of temperature and precipitation differences (or changes) on NEE. Zhang et al. (2012) is one of the few studies that analyzed in more detail the role of uncertain meteorological forcings (together with parameter uncertainty) for calculating NEE with a process-based ecosystem model. Their simulations for a Korean pine mixed forest site with two different input meteorological datasets revealed clear differences in model response. Spadavecchia et al. (2011) performed a rigorous uncertainty analysis for the combination of parameter and forcing

uncertainty with a simple LSM for a pine stand in Oregon, USA. They found that the contribution of parameter uncertainty to NEE-uncertainty is larger than the contribution of forcing uncertainty, but forcing uncertainty also has a significant impact. If meteorological stations are located more than 100km away from the study site, forcing uncertainty starts to dominate over parameter uncertainty. The relative contribution of initial state uncertainty to prediction uncertainty of LSMs has not been subject of intensive study although it was recognized as one of the major challenges in model-data fusion studies (Williams et al., 2009). Richardson et al. (2010) performed a Monte Carlo type inverse modelling to assimilate many different data types in a forest carbon cycle model for the Howland AmeriFlux-site in Maine, USA. They estimated jointly model parameters and initial carbon pools. Although they found that model performance improved strongly, they also concluded that carbon pools are more difficult to constrain than ecosystem parameters. Peylin et al. (2016) also updated jointly parameters and carbon pools, in this case with the land surface model ORCHIDEE for a large scale application. These last two studies explicitly considered initial state uncertainty together with other uncertainty sources, but did not focus on the relative contribution of the different uncertainty sources. As considering atmospheric forcings and initial states uncertainty, besides parameter uncertainty, has not received so much attention in the literature until now, we investigate the uncertainty of model predictions in relation to uncertain model parameters, initial conditions and atmospheric forcings in this work ."

p3 l25: see also Xia et al. (2012) / Exbrayat et al. (2014a);

These references have been added here as further examples in the literature where uncertainty in the initial conditions is addressed.

p3 l27: please consider adding a reference to the Global Carbon Budget TRENDY intercomparison project here

A reference (Zhao et al., 2016) has been added.

p5 l2: please indicate time series length here

Time series lengths are summarized in Table 1. This is indicated now in the modified sentence: "Eddy covariance (EC) data in the Rur catchment were measured four C3-crop sites for the evaluation periods summarized in Table 1."

p5 l10-12: although this has been reported in a previous paper, can you please develop a bit more on how parameters were optimized for these sites?

In section 2.3.1 we explained how parameters were optimized and refer to the previous paper for further details. As the focus of this paper is not on parameter estimation but on model uncertainty (which we will now better highlight in the abstract), we would not extent the outline of the parameter estimation methods. If there is some particular information you are missing, we can add it.

p6 l13: can you indicate the respective fraction of each PFT here instead of l25

The sentence "The percentage PFT coverage of the vegetated land in Rur catchment was ~34% C3-crops, ~32% grassland, ~17% broadleaf deciduous forest and ~14% coniferous forest." was moved upwards as suggested.

p7 l10: how representative of mean climate were the three years used for spin-up?

The following text was added for clarification:

"According to http://www.dwd.de/DE/klimaumwelt/klimaatlas/klimaatlas_node.html, the annual average temperature in the years 2008 and 2009 was ~ 0.5-1.0 °C higher than the long term average (1961-1990) and the mean annual temperature in 2010 was ~ 0.5-1.0 °C lower. Mean precipitation amounts and freezing days were representative for the long term average.

We also studied the effect of the forcing data used for the model spin-up. For example, we tested using a longer time series (1998-2004) of global climate data. However, we found that using more recent, regional forcing data during the spin-up resulted in carbon- and energy fluxes in better agreement with the observations."

p7 l11: what is an "exit" spin-up? does that simply mean that the spin-up was 1203 years repeated meteorological drivers for 2008-2010

Principally, yes. We reformulated this sentence. "Exit spin-up" is a technical term for CLM. Before conducting the actual model runs with the restart files from the model spin-up, an intermediate run (the exit spin-up") is necessary with specific parameter settings. This is e.g. because CLM uses an accelerated model spin-up mode. In the paper is clarified (from P8 L11):

"The model states obtained after the 1200-year spin-up were then used as input for a second three years "exit spin-up" also using the meteorological data for the years 2008-2010. The "exit spin-up" in CLM is necessary for technical reasons and switches the CLM settings from the (accelerated) spin-up mode to the "normal" mode in terms of the calculated carbon-nitrogen cycling."

p7 l24: what value of the convergence diagnostic did you use?

We used a threshold of 1.2 to declare convergence as suggested e.g. in Vrugt (2015). This information is added now:

"Convergence is achieved if $\hat{R}$ is smaller than 1.2 for all parameters."

p8 l9: please add reference for Fontainebleau's site

We added a reference from the Fontainebleau team. However, we did not find a specific paper describing the site but a large number of papers where data from the site are used.

p8 l26: compared

It is not clear what the reviewer means. In this sentence we used the word "compared" and think this is correct.

p10 l1: please add full name and references for the two sites.

Full name and references for the two sites have been added.

p11 l19: are 15 years enough for this perturbed spin-up?

We think so. We also tested longer perturbed spin-up periods up to 30 years and found there were no significant differences of the ensemble spread. We found that after about 8 years, the ensemble spread reaches some kind of maximum and then does not increase considerably further. However, at the same time we already commented in the paper:

"This perturbation takes into account the uncertainty of the less stable carbon and nitrogen pools, but uncertainty with respect to the stable and resistant carbon and nitrogen pools is not taken into account."

p12 l8: I am not clear what "deterministic initial states" mean here

We changed this sentence and added another sentence for clarification:

"CLM-Ens$_P$: ensemble model runs for 1 Dec 2012 – 30 Nov 2013 with deterministic (non-perturbed) initial states and non-perturbed input data from COSMO-DE (Sect.2.2). The initial states are the outcome of the default 1203 years spin-up, without perturbations."

p13 l1: why are results for ENSp outside the range from ENSpai? from the experiment description it feels that ENSpai's uncertainty should envelop that of ENSp (and ENSpa)

This is possible because the initial states are not the same. Because parameters and initial states are strongly dependent on each other and because initial states have changed after the perturbed spin up, the „reaction" of the model to the parameters is different. It might also show a model sensitivity which is too high. Here just the model response is reported. It was however checked that there was no mistake in the performed simulations.

p13 l21: there is a problem with figure numbering.

The figure numbering has been corrected.

p13 l23: daytime NEE is not GPP, please replace/revise in the rest of the paragraph.

Thanks, this is true. This text block was modified to:

"The delay of the plant emergence indicated by these LAI courses is related to the strong underestimation of daytime NEE in spring 2013 (**Error! Reference source not found.**, **Error! Reference source not found.**). This underestimation of NEE can mainly be attributed to an underestimation of GPP, which is too low in spring because the simulated plant onset was about 2 weeks later than observed. For CLM-Ref and for most of the CLM-Ens$_{PAI}$ ensemble members, leaf onset started in May. For those model realizations, the underestimation of daytime NEE in spring was highest. In contrast, for CLM-Ens$_P$ and a small proportion of CLM-Ens$_{PAI}$, leaf onset already started in March. For those cases, the underestimation of daytime NEE in spring was notably lower."

p 15 l17: the discussion is very short in regards of the amount of results that are described.

We extended a bit the part related to model-data consistency for the estimated parameters. The added text part is (from P17 L16):

"Therefore, we expect that the performance of CLM can be improved if it is coupled to a crop model, which can treat specific plant traits and include management practices. Li et al. (2011) already found for the coupled LSM-crop model ORCHIDEE-STICS that crop variety and management factors like irrigation, fertilization and planting date have a significant impact on NEE (and evapotranspiration). Wu et al. (2016) showed that a crop model coupled to the LSM ORCHIDEE, combined with assimilation of many different data types, was able to reproduce many measurement data up to the measurement uncertainty, whereas other remaining errors could be related to management activities which were not correctly represented in the model."

We extended considerably the discussion on model uncertainty (from P17 L22):

"The simulation experiments that considered uncertainty in meteorological forcings, initial states and parameters, together with an additional experiment that considered uncertainty in $Q_{10}$, gave for the forest PFTs results in correspondence with expectations. Uncertainty of the $Q_{10}$-parameter had the largest impact on NEE, followed by uncertainty of the initial states and uncertainty of meteorological forcings. It is important to stress again that uncertainty in initial states was considered by an additional 15 years spin-up with perturbed parameters and meteorological forcings. This short additional spin-up does not

consider uncertainty of the most stable carbon and nitrogen pools. Carbon and nitrogen pools are also influenced by changing land use in the last centuries and management practices, which are also not considered. Therefore, the true uncertainty related to the initial states will be larger than in these simulation experiments. The simulation results for C3-grass and C3-crop were surprising as uncertainty in the meteorological forcings had a much larger impact on uncertainty of NEE than the uncertainty of initial states and $Q_{10}$ had. We argue that the applied meteorological perturbations (standard deviations of 1.0K for air temperature, 30% of the incoming shortwave radiation and 50% of precipitation amount) are realistic for meteorological reanalysis products, and in correspondence with other studies. The perturbations (which are also temporally correlated) result in considerable variations, between ensemble members, of leaf onset and senescence. The stress deciduous phenology scheme (which determines plant onset and offset for C3-crop and C3-grass in CLM as outlined in Section 2.2) is strongly determined by various arbitrary thresholds such as "crit_offset_swi" (water stress days for offset trigger) or "crit_onset_fdd" (critical number of freezing degree days to trigger onset). The phenology representation of C3-crop and C3-grass could be too sensitive to those thresholds (Dahlin et al. 2015). Verheijen et al. (2015) argue that the use of PFT´s instead of plant traits in LSMs reduces the adaptive response of vegetation to environmental drivers. A second important reason for the large impact of the perturbation of meteorological forcings on NEE-uncertainty is the preferential location of the C3-crop and C3-grass in the drier, northern part of the catchment, which experiences during summer some drought stress. The perturbation of the precipitation amounts affects strongly this drought stress, resulting in a large variability in NEE and LAI. This shows that the impact of the uncertainty of meteorological forcings depends on the local conditions. Jung et al. (2011) already showed that interannual variability of NEE is larger in semi-arid and semi-humid areas than in other regions. In summary, the large impact of uncertainty of meteorological input on NEE-uncertainty is likely partly model specific, but also related to the semi-humid conditions in the agricultural part of the region. High impact of uncertainty of meteorological conditions can be expected in other semi-humid and semi-arid regions and should be taken into account in simulation studies.

In order to better estimate the uncertainty of the initial carbon and nitrogen pools, it would be necessary to include the spin-up (for each of the ensemble members) in the data assimilation experiments where parameters are estimated. As model runs have to be repeated thousands of times, this would be extremely CPU-expensive. The spin-up should also consider uncertainty of historical land use and uncertainty of historical meteorological conditions. Especially the land use and meteorological conditions of the last few centuries would impact the initial states. We believe that it is important to consider jointly the uncertainty of initial conditions and parameters in LSM forward and inverse modelling runs. Pinnington et al. (2016) already pointed towards the importance of considering correlations between initial states and parameters in inverse modelling studies."

Figures 3, 4 and 5: please correct the legend

The legends have been corrected. See the new figures.

**REVIEWER 2**

Thank you for positive feedback and suggestions to improve the paper. We considered all suggestions as prescribed below.

I find this a very interesting and robust study, which unfortunately still needs a bit of fine tuning, in particular in the presentation. By reading the abstract I was under the impression that this study was "merely" an application of a data assimilation method result to larger scales, whereas in truth the study expands to uncertainty in met-forcing and initial conditions, which is typically ignored in large-scale applications. I think this achievement deserves a bit more presentation in abstract, results and also discussion.

Thanks. The abstract has been revised. The further suggestions to improve also introduction, results and discussion are handled as well.

In the introduction, the text could be streamlined to focus on the current study and avoid side-lines as the use of atmospheric inversions or the future predicted climate of the Rur valley.

We streamlined the introduction and deleted a paragraph related to atmospheric inversion and the future predicted climate of the Rur valley. In addition, the introduction is further modified as pointed out in the answers given to your comments below.

I've been missing an introduction section that addresses in some more details the motivation for not only using DA to obtain model parameters, but at the same time use of perturbed boundary conditions (meteorological and soil), and a discussion of this in Section 4. I think this is really a novel contribution, which deserves somewhat more place.

The introduction has been modified, and especially the motivation for considering uncertainty in initial states and meteorological forcings has been strengthened (from P3, L16):

"Some studies estimate the uncertainty of terrestrial carbon flux predictions based on an ensemble of many different LSMs (e.g. Fisher et al., 2014; Huntzinger et al., 2012; Piao et al., 2013; Zhao et al., 2016). These studies focus on differences between models and therefore model structural deficits. Those studies highlight that (i) carbon flux predictions are generally highly uncertain, which also contributes to the uncertainty in climate change predictions, (ii) interactions of the different processes and drivers is not understood satisfactorily, and (iii) models require structural improvement to produce more consistent predictions. In order to improve LSM model structure and thus model-data and inter-model consistency, a more comprehensive understanding of model functionality and the contribution and link of the different model error sources is required. However, as highlighted by Xiao et al. (2014), the uncertainty of carbon fluxes obtained by ecosystem and land surface models has not been analyzed and quantified enough, particularly in regional scale studies.

Whereas the uncertainty of land surface model parameters has been subject to intensive investigations (e.g., Ren et al., 2013; Xiao et al., 2014), and several works are dedicated to reducing this uncertainty, for example by parameter estimation with data assimilation methods (e.g., Safta et al., 2015), the other sources of uncertainty are less studied. It was already concluded in earlier work that parameter uncertainty alone cannot explain observed deviations between measured and simulated NEE (e.g., Pridhoko et al., 2008; Wang et al., 2011), and the remaining deviations are often attributed to model structural errors. However, also uncertainty in atmospheric forcings and initial conditions could contribute to unexplained deviations between simulated and measured NEE. From global scale studies it is known that NEE is sensitive to the climate scenario. For example, in a study the LSM LPJ-GUESS was forced with output from 18 different coupled atmosphere-ocean global circulation models or earth system models, and showed very different NEE-responses depending on the climate scenario. Ten out of 18 models project that the land will become a carbon source in the 21st century, while the other eight models indicate that the land will act as a carbon sink (Ahlstrom et al., 2012). At the plot, catchment or regional scale and for shorter

time periods in the recent past, very few studies analyzed the impact of uncertainty in meteorological forcings on NEE. Studies which analyzed interannual variability in NEE detected the important role of temperature and precipitation as drivers of this variability (e.g., Keppel-Aleks et al., 2014). Gu et al. (2016) found that for the Missouri Ozark AmeriFlux forest site, the interannual NEE-variability is smaller in simulations by CLM than in the data. On the other hand, some field studies for pairs of monitoring sites (Kwon et al., 2006) or experimentation sites (with temperature increase and precipitation increase or decrease) (Xu et al., 2016) found a limited impact of temperature and precipitation differences (or changes) on NEE. Zhang et al. (2012) is one of the few studies that analyzed in more detail the role of uncertain meteorological forcings (together with parameter uncertainty) for calculating NEE with a process-based ecosystem model. Their simulations for a Korean pine mixed forest site with two different input meteorological datasets revealed clear differences in model response. Spadavecchia et al. (2011) performed a rigorous uncertainty analysis for the combination of parameter and forcing uncertainty with a simple LSM for a pine stand in Oregon, USA. They found that the contribution of parameter uncertainty to NEE-uncertainty is larger than the contribution of forcing uncertainty, but forcing uncertainty also has a significant impact. If meteorological stations are located more than 100km away from the study site, forcing uncertainty starts to dominate over parameter uncertainty. The relative contribution of initial state uncertainty to prediction uncertainty of LSMs has not been subject of intensive study although it was recognized as one of the major challenges in model-data fusion studies (Williams et al., 2009). Richardson et al. (2010) performed a Monte Carlo type inverse modelling to assimilate many different data types in a forest carbon cycle model for the Howland AmeriFlux-site in Maine, USA. They estimated jointly model parameters and initial carbon pools. Although they found that model performance improved strongly, they also concluded that carbon pools are more difficult to constrain than ecosystem parameters. Peylin et al. (2016) also updated jointly parameters and carbon pools, in this case with the land surface model ORCHIDEE for a large scale application. These last two studies explicitly considered initial state uncertainty together with other uncertainty sources, but did not focus on the relative contribution of the different uncertainty sources. As considering atmospheric forcings and initial states uncertainty, besides parameter uncertainty, has not received so much attention in the literature until now, we investigate the uncertainty of model predictions in relation to uncertain model parameters, initial conditions and atmospheric forcings in this work. "

The discussion was also modified to accommodate better the analysis of the impact of uncertainty in forcings and initial states, which is highlighted in one of the next questions by the reviewer.

Methodologically, I am unsure whether the assimilation procedure included the spin-up period, or not? In the latter case, one should caution the interpretation of the effect of the assimilation on the cumulative NEE?

The assimilation procedure (where parameters were estimated) did not include a spin-up period. Additional clarification is given (from P9 L31):

"(ii) An ensemble run with 60 realizations, for the same time period, same initial conditions and same atmospheric input data as CLM-Ref, but with parameters sampled randomly from the DREAM multivariate posterior pdfs (CLM-Ens$_P$). This implies that we did not perform a separate model spin-up for each of the estimated set of parameters, as this was computationally too expensive."

As shown e.g. in Post et al. (2016), parameters strongly depend on the set of initial conditions. A main conclusion of this paper was that strictly speaking, estimated CLM parameters cannot necessarily be considered valid when combined with initial states very different from the states that were used during parameter estimation and model evaluation. Here, we should use the estimated parameters with the same initial states that were used for the parameter estimation, in order to achieve an optimal performance. The strong correlation of initial states like carbon-nitrogen pools and parameters is a central problem in the current usage of complex land surface models. Unfortunately, due to the high computational demand, it is not possible to include the spin-up in the parameter estimation procedure for land surface models like CLM.

The discussion is simply too cursory and needs to better take account of existing literature and explaining the added insights gained from this study. This could be partially achieved by taking considerations from the conclusions and expanding these ideas in the light of the existing literature (or potential applications for regional modelling).

We extended a bit the part related to model-data consistency for the estimated parameters. The added text part is (from P17 L16):

"Therefore, we expect that the performance of CLM can be improved if it is coupled to a crop model, which can treat specific plant traits and include management practices. Li et al. (2011) already found for the coupled LSM-crop model ORCHIDEE-STICS that crop variety and management factors like irrigation, fertilization and planting date have a significant impact on NEE (and evapotranspiration). Wu et al. (2016) showed that a crop model coupled to the LSM ORCHIDEE, combined with assimilation of many different data types, was able to reproduce many measurement data up to the measurement uncertainty, whereas other remaining errors could be related to management activities which were not correctly represented in the model."

We extended considerably the discussion on model uncertainty (from P17 L22):

"The simulation experiments that considered uncertainty in meteorological forcings, initial states and parameters were considered, together with an additional experiment that considered uncertainty in $Q_{10}$, gave for the forest PFTs results in correspondence with expectations. Uncertainty of the $Q_{10}$-parameter had the largest impact on NEE, followed by uncertainty of the initial states and uncertainty of meteorological forcings. It is important to stress again that uncertainty in initial states was considered by an additional 15 years spin-up with perturbed parameters and meteorological forcings. This short additional spin-up does not consider uncertainty of the most stable carbon and nitrogen pools. Carbon and nitrogen pools are also influenced by changing land use in the last centuries and management practices, which are also not considered. Therefore, the true uncertainty related to the initial states will be larger than in these simulation experiments. The simulation results for C3-grass and C3-crop were surprising as uncertainty in the meteorological forcings had a much larger impact on uncertainty of NEE than the uncertainty of initial states and $Q_{10}$ had. We argue that the applied meteorological perturbations (standard deviations of 1.0K for air temperature, 30% of the incoming shortwave radiation and 50% of precipitation amount) are realistic for meteorological reanalysis products, and in correspondence with other studies. The perturbations (which are also temporally correlated) result in considerable variations, between ensemble members, of leaf onset and senescence. The stress deciduous phenology scheme (which determines plant onset and offset for C3-crop and C3-grass in CLM as outlined in Section 2.2) is strongly determined by various arbitrary thresholds such as "crit_offset_swi" (water stress days for offset trigger) or "crit_onset_fdd" (critical number of freezing degree days to trigger onset). The phenology representation of C3-crop and C3-grass could be too sensitive to those thresholds (Dahlin et al. 2015). Verheijen et al. (2015) argue that the use of PFT´s instead of plant traits in LSM´s reduces the adaptive response of vegetation to environmental drivers. A second important reason for the large impact of the perturbation of meteorological forcings on NEE-uncertainty is the preferential location of the C3-crop and C3-grass in the drier, northern part of the catchment, which experiences during summer some drought stress. The perturbation of the precipitation amounts affects strongly this drought stress, resulting in a large variability in NEE and LAI. This shows that the impact of the uncertainty of meteorological forcings depends on the local conditions. Jung et al. (2011) already showed that interannual variability of NEE is larger in semi-arid and semi-humid areas than in other regions. In summary, the large impact of uncertainty of meteorological input on NEE-uncertainty is likely partly model specific, but also related to the semi-humid conditions in the agricultural part of the region. High impact of uncertainty of meteorological conditions can be expected in other semi-humid and semi-arid regions and should be taken into account in simulation studies.

In order to better estimate the uncertainty of the initial carbon and nitrogen pools, it would be necessary to include the spin-up (for each of the ensemble members) in the data assimilation experiments where parameters are estimated. As model runs have to be repeated thousands of times, this would be extremely CPU-expensive. The spin-up should also consider uncertainty of historical land use and uncertainty of historical meteorological conditions. Especially the land use and

meteorological conditions of the last few centuries would impact the initial states. We believe that it is important to consider jointly the uncertainty of initial conditions and parameters in LSM forward and inverse modelling runs. Pinnington et al. (2016) already pointed towards the importance of considering correlations between initial states and parameters in inverse modelling studies."

Finally, also the conclusions were reformulated in the light of some new insight and this paragraph is mostly new:

"The forest PFTs were less sensitive to uncertainty of atmospheric forcings than C3-grass and C3-crop. The model uncertainty resulting from uncertain atmospheric forcings for C3-grass and C3-crop is enhanced as these crops are located in the northern part of the catchment which is prone to some drought stress in summer, which was enhanced or reduced for the different ensemble members depending on the perturbation of the precipitation, resulting in large NEE and LAI variations. In addition, leaf onset and senescence showed a considerable variation among ensemble members, which might be related to the phenology representation in CLM which is too sensitive to certain thresholds. Improving possible model deficits in the future is of high relevance for obtaining more reliable carbon fluxes and carbon stock predictions, particularly in terms of their response to climate change."

**Minor comments:**

The abstract is quite long, consider shortening.

The abstract has been shortened by about 20% with less focus on details of the model-data fusion and more focus on the uncertainty of modelled NEE.

P1 L19: give average error reduction in mol CO2 / m2 / s ?

We did not include absolute error reductions here, because we used non gap filled data to calculate the NEE sum. The absolute values could be misleading.

p1 L21: a) is this in agreement with the observations (the fact that the NEE goes positive); b) add "simulated" before regional carbon balance estimates

    a)   Yes. As shown in Figure 2, EC data indicated that the NEE sum was much smaller (more negative) than the simulated NEE sum. This is mainly due to an underestimation of LAI and GPP in spring with default parameters, as discussed in the last paragraph of section 3.1. Given that the different EC sites are representative for large parts of the catchment area, we assume that CLM estimates of the catchment scale NEE sum are more reliable with estimated parameters than with default parameters. We added "(and closer to observations)" to the sentence.

    b)   "simulated" is added as suggested.

P1 L22: here and elsewhere it would be important to understand whether you have brought the model into equilibrium in terms of the carbon cycle with the parameter set used, or whether you relied on a different way to initialise carbon stocks

This was shortly mentioned in the paper, but now we add an additional sentence to stress this (from P9 L31):

"(ii) An ensemble run with 60 realizations, for the same time period, same initial conditions and same atmospheric input data as CLM-Ref, but with parameters sampled randomly from the DREAM multivariate posterior pdfs (CLM-Ens$_P$). This implies that we did not perform a separate model spin-up for each of the estimated set of parameters, as this was computationally too expensive."

As shown e.g. in Post et al. (2016), parameters strongly depend on the set of initial conditions. A main conclusion of this paper was that strictly speaking, estimated CLM parameters cannot necessarily be considered valid when combined with

initial states very different from the states that were used during parameter estimation and model evaluation. Here, we should use the estimated parameters with the same initial states that were used for the parameter estimation, in order to achieve an optimal performance. The strong correlation of initial states like carbon-nitrogen pools and parameters is a central problem in the current usage of complex land surface models. Unfortunately, due to the high computational demand, it is not possible to include the spin-up in the parameter estimation procedure for land surface models like CLM.

P1 L26: It would be helpful to add in brackets the difference in annual integrated NEE in mol / m2 / year

This information has been added and the sentence is now (from P1 L20):

"C3-grass and C3-crop were particularly sensitive to the perturbed meteorological input, which resulted in a strong increase of the standard deviation of the NEE sum ($\sigma_{\sum NEE}$) for the different ensemble members from ~2-3 gC m$^{-2}$ y$^{-1}$ (with uncertain parameters) to ~45 gC m$^{-2}$ y$^{-1}$ (C3-grass ) and ~75 gC m$^{-2}$ y$^{-1}$ (C3-crop) with perturbed forcings and initial states."

P1 L29: If the uncertainty was indeed reduced, then this would need to be demonstrated by comparing prior and posterior distributions of the regional NEE. I guess you rather would like to state that the accuracy of the projection has increased because the model better fits site-level observations?

Although we found, by comparing prior and posterior pdf´s, that the uncertainty of NEE which can be attributed to parameter uncertainty indeed decreased, we would not like to focus on this. The paper wants to stress that it is important to take also uncertainty with respect to initial states and meteorological forcings into account. The text has been reformulated and we skip now the reference to the uncertainty reduction by parameter estimation. Please have a look at the new abstract.

P2 L11: define what "conventional interpolation methods" are

We added:

"like for example kriging or inverse distance weighting"

P2 L12ff: all correct, but I don't think this paragraph is necessary for the sake of this paper.

We agree and deleted this complete paragraph.

P3 L2: please define more precisely what "error" is here. There is no principle error in averaging the input data, and then obtaining model output from this. The question is whether the aggregation method is sufficiently representing the average regional flux, is of course relevant.

Indeed, this formulation was not precise enough. We now write (from P2 L26):

"With such a high degree of spatial aggregation, and given the non-linearity of the governing equations, the modelled values for output variables like NEE at this coarse scale can deviate strongly from the ones that would have been calculated with a fine resolution model using fine resolution input."

P3 L32: I disagree that the uncertainty of carbon fluxes has been overlooked so far. I agree that the uncertainty hasn't been sufficiently quantified, and/or reduced.

This particular sentence was slightly reformulated, also given literature published in the last few years (from P3 L23):

"However, as highlighted by Xiao et al. (2014), the uncertainty of carbon fluxes obtained by ecosystem and land surface models has not been analyzed and quantified enough, particularly in regional scale studies. "

P4L1: well, if you counted conference papers, it actually has. I don't think the question of "who was first" is really relevant, and would focus more on the value and design of the regional set-up

The introduction was reformulated and this particular sentence has been skipped. We motivate why we analyzed the role of forcing and initial state uncertainty and argue that these sources of uncertainty are often not considered. See the text block which was cited in response to one of your major comments.

P4 L6: define what a "validated parameter" is. I assume that this is a parameter derived from data-assimilation? It is unclear to me why new estimates of parameters have been obtained for one PFT. Is this because you extended your PFT set to a new PFT (from 2016), or because the DA method of 2016 did not yield good parameters. In the latter case, would it not have been appropriate to recalibrate the model for all sites?

"Validated parameter estimates" refers to the parameter estimates which have been successfully estimated and evaluated in the previous study, i.e. yielded NEE model outputs closer to the measured values than the default CLM parameter values. For one PFT (C3-crop) the estimated parameter values from the previous study could not be successfully evaluated, i.e. did not clearly reduce the model-data misfit. Therefore, we estimated new parameter values for the C3-crop in this study. This is better clarified now in the text:

"For C3-crop, we estimated a new set of parameters as the estimated parameters by Post et al. (2016) did not improve the model-data fit in verification experiments, using the DiffeRential Evolution Adaptive Metropolis DREAM (Vrugt, 2015)."

P6 L24: add "average" before percentage PFT cover.

This has been added now and the sentence has been moved in correspondence with a comment by reviewer #1.

P7 L11: Please briefly explain, why it was necessary to add a second spin-up

This is the default procedure suggested in the CLM user manual and is mainly necessary due to technical reasons. This is clarified in the paper as follows (from P8 L11):

"The model states obtained after the 1200-year spin-up were then used as input for a second three years "exit spin-up" also using the meteorological data for the years 2008-2010. The "exit spin-up" in CLM is necessary for technical reasons and switches the CLM settings from the (accelerated) spin-up mode to the "normal" mode in terms of the calculated carbon-nitrogen cycling."

P7 L22: The statement of robustness of the method either requires a proof or (more likely) a reference

A reference has been added here (Vrugt, 2015).

P11 L 25: I make this note here, although this probably needs mentioning later in that by not manipulating the "stable" nitrogen pool you basically conserve the amount of N available from net N mineralisation across the ensemble. That's fine (in particular, because the system are fertilised), but probably implies that you underestimate the true uncertainty in the net N availability at the sites.

We agree and have added this in the text (from P13 L1):

"As the stable nitrogen pool is not perturbed, the availability of N from mineralization is very similar across the ensemble members. This might be an unproblematic assumption given N fertilization by agriculture and atmospheric deposition, but also implies that the uncertainty of the net N availability at the sites is likely underestimated."

P15 L7 (and elsewhere): please refer to other parts of the manuscript as sections, not chapters.

This has been modified (there was only one occurrence in the paper).

P15L11: Please be more precise, this is not really a general finding, it is specific to Q10. Please have the revised manuscript cross-checked by a native speaker, and remove unnecessary words (such as "however" in the middle of a sentence, e.g. P15 L8).

One single key parameter has been replaced by Q10. The sentence is now (from P16 L20):

"This highlights how the uncertainty of $Q_{10}$ can affect the uncertainty of predicted carbon fluxes in CLM."

P15: This discussion section is a bluntly speaking a bit too skinny. More could be made for instance of the extremely relevant and very nicely demonstrated point that minor modifications in input meteorology lead to markable differences in modelled NEE.

We worked on this and added also some new insights. We repeat here again the text parts which have been added to the discussion (from P17 L16):

"Therefore, we expect that the performance of CLM can be improved if it is coupled to a crop model, which can treat specific plant traits and include management practices. Li et al. (2011) already found for the coupled LSM-crop model ORCHIDEE-STICS that crop variety and management factors like irrigation, fertilization and planting date have a significant impact on NEE (and evapotranspiration). Wu et al. (2016) showed that a crop model coupled to the LSM ORCHIDEE, combined with assimilation of many different data types, was able to reproduce many measurement data up to the measurement uncertainty, whereas other remaining errors could be related to management activities which were not correctly represented in the model."

We extended considerably the discussion on model uncertainty (from P17 L22):

"The simulation experiments that considered uncertainty in meteorological forcings, initial states and parameters were considered, together with an additional experiment that considered uncertainty in $Q_{10}$, gave for the forest PFTs results in correspondence with expectations. Uncertainty of the $Q_{10}$-parameter had the largest impact on NEE, followed by uncertainty of the initial states and uncertainty of meteorological forcings. It is important to stress again that uncertainty in initial states was considered by an additional 15 years spin-up with perturbed parameters and meteorological forcings. This short additional spin-up does not consider uncertainty of the most stable carbon and nitrogen pools. Carbon and nitrogen pools are also influenced by changing land use in the last centuries and management practices, which are also not considered. Therefore, the true uncertainty related to the initial states will be larger than in these simulation experiments. The simulation results for C3-grass and C3-crop were surprising as uncertainty in the meteorological forcings had a much larger impact on uncertainty of NEE than the uncertainty of initial states and $Q_{10}$ had. We argue that the applied meteorological perturbations (standard deviations of 1.0K for air temperature, 30% of the incoming shortwave radiation and 50% of precipitation amount) are realistic for meteorological reanalysis products, and in correspondence with other studies. The perturbations (which are also temporally correlated) result in considerable variations, between ensemble members, of leaf onset and senescence. The stress deciduous phenology scheme (which determines plant onset and offset for C3-crop and C3-grass in CLM as outlined in Section 2.2) is strongly determined by various arbitrary thresholds such as "crit_offset_swi" (water stress days for offset trigger) or "crit_onset_fdd" (critical number of freezing degree days to trigger onset). The phenology representation of C3-

crop and C3-grass could be too sensitive to those thresholds (Dahlin et al. 2015). Verheijen et al. (2015) argue that the use of PFT´s instead of plant traits in LSM´s reduces the adaptive response of vegetation to environmental drivers. A second important reason for the large impact of the perturbation of meteorological forcings on NEE-uncertainty is the preferential location of the C3-crop and C3-grass in the drier, northern part of the catchment, which experiences during summer some drought stress. The perturbation of the precipitation amounts affects strongly this drought stress, resulting in a large variability in NEE and LAI. This shows that the impact of the uncertainty of meteorological forcings depends on the local conditions. Jung et al. (2011) already showed that interannual variability of NEE is larger in semi-arid and semi-humid areas than in other regions. In summary, the large impact of uncertainty of meteorological input on NEE-uncertainty is likely partly model specific, but also related to the semi-humid conditions in the agricultural part of the region. High impact of uncertainty of meteorological conditions can be expected in other semi-humid and semi-arid regions and should be taken into account in simulation studies.

In order to better estimate the uncertainty of the initial carbon and nitrogen pools, it would be necessary to include the spin-up (for each of the ensemble members) in the data assimilation experiments where parameters are estimated. As model runs have to be repeated thousands of times, this would be extremely CPU-expensive. The spin-up should also consider uncertainty of historical land use and uncertainty of historical meteorological conditions. Especially the land use and meteorological conditions of the last few centuries would impact the initial states. We believe that it is important to consider jointly the uncertainty of initial conditions and parameters in LSM forward and inverse modelling runs. Pinnington et al. (2016) already pointed towards the importance of considering correlations between initial states and parameters in inverse modelling studies."

However, generally, a critical assessment of the novelty of the study approach and finding compared to the existing literature is missing.

The literature review and motivation was extended as pointed out earlier. The introduction was extended considerably. Few studies were performed which can be directly compared to this study, but we were able to explain the large impact of the perturbation of meteorological forcings on NEE-uncertainty.

Table 3: Please arrange the table such that it is immediately clear which CLM4.5 default simulation belongs to which site

The table has been re-organized to clarify this.

Table 4: please briefly explain the abbreviations used to denote "grid cells", and how the "n" were calculated (or what they are).

Explanations have been added in the figure caption (Table 4):

"Results are given for ME (Merzenhausen), SE (Selhausen), NZ (Niederzier) and EN (Engelskirchen). Here, $n$ is the amount of non-gap filled half-hourly measurement data that was available to calculate these evaluation indices."

Table 5: define "n"

We changed the used symbol to $N$. the following explanation was added (Caption Table 5):

"$N$ is the number of grid cells in the catchment covered by more than 80% by one specific PFT."

Figure 2: define "evaluation period". It would also be preferable to somewhat make a clearer distinction between different model ensembles and the observations

The figure caption has been modified to indicate that this information can be found in Table 1. As the evaluation periods differ among the sites, we feel that it is better not to repeat everything again in the caption.

[revised manuscript text omitted]

~~The two central approaches to obtain spatially distributed carbon flux estimates for larger areas are either (i) inverse atmospheric modeling approaches (Deng et al., 2007; Peters et al., 2007; Peylin et al., 2013; Chevallier et al., 2014), or (ii) the application of terrestrial ecosystem models or land surface models (LSMs) such as the Community Land Model CLM (Oleson et al., 2013). In case of inverse atmospheric modeling, atmospheric transport models are combined with observed atmospheric CO₂ concentrations, using data assimilation methods such as the Ensemble Kalman Filter (e.g. Peters et al.,~~

2007; Tolk et al., 2011). Inverse atmospheric modeling mainly provides carbon flux estimates at continental or global scales and coarse spatial resolutions (Deng et al., 2007), whereas flux estimates at regional scales are highly biased (Tolk et al., 2011; Chevallier et al., 2014). As highlighted in Turner et al. (2011) such "top-down" approaches are not suited to estimate small scale patterns of terrestrial carbon fluxes.

[revised manuscript text omitted]

time periods in the recent past, very few studies analyzed the impact of uncertainty in meteorological forcings on NEE. Studies which analyzed interannual variability in NEE detected the important role of temperature and precipitation as drivers of this variability (e.g., Keppel-Aleks et al., 2014). Gu et al. (2016) found that for the Missouri Ozark AmeriFlux forest site, the interannual NEE-variability is smaller in simulations by CLM than in the data. On the other hand, some field studies for pairs of monitoring sites (Kwon et al., 2006) or experimentation sites (with temperature increase and precipitation increase or decrease) (Xu et al., 2016) found a limited impact of temperature and precipitation differences (or changes) on NEE. Zhang et al. (2012) is one of the few studies that analyzed in more detail the role of uncertain meteorological forcings (together with parameter uncertainty) for calculating NEE with a process-based ecosystem model. Their simulations for a Korean pine mixed forest site with two different input meteorological datasets revealed clear differences in model response. Spadavecchia et al. (2011) performed a rigorous uncertainty analysis for the combination of parameter and forcing uncertainty with a simple LSM for a pine stand in Oregon, USA. They found that the contribution of parameter uncertainty to NEE-uncertainty is larger than the contribution of forcing uncertainty, but forcing uncertainty also has a significant impact. If meteorological stations are located more than 100km away from the study site, forcing uncertainty starts to dominate over parameter uncertainty. The relative contribution of initial state uncertainty to prediction uncertainty of LSMs has not been subject of intensive study although it was recognized as one of the major challenges in model-data fusion studies (Williams et al., 2009). Richardson et al. (2010) performed a Monte Carlo type inverse modelling to assimilate many different data types in a forest carbon cycle model for the Howland AmeriFlux-site in Maine, USA. They estimated jointly model parameters and initial carbon pools. Although they found that model performance improved strongly, they also concluded that carbon pools are more difficult to constrain than ecosystem parameters. Peylin et al. (2016) also updated jointly parameters and carbon pools, in this case with the land surface model ORCHIDEE for a large scale application. These last two studies explicitly considered initial state uncertainty together with other uncertainty sources, but did not focus on the relative contribution of the different uncertainty sources. Some studies estimate the uncertainty of terrestrial carbon flux predictions based on an ensemble or comparisons of many different LSMs (e.g. Fisher et al., 2014; Huntzinger et al., 2012; Piao et al., 2013). These studies highlight that (i) carbon flux predictions are generally highly uncertain, which also contributes to the uncertainty in climate change predictions, (ii) interactions of the different processes and drivers is not understood satisfactorily, and (iii) models require structural improvement to produce more consistent predictions. In order to improve LSM model structure and thus model data and inter-model consistency, a more comprehensive understanding of model functionality and the contribution and link of the different model error sources is required. However, as highlighted by Xiao et al. (2014), the uncertainty of carbon fluxes obtained by ecosystem models has largely been overlooked, particularly in regional scale studies. The same is true for land surface models. As considering atmospheric forcings and initial states uncertainty, besides parameter uncertainty, has not received so much attention in the literature until now, we investigate the uncertainty of model predictions in relation to uncertain model parameters, initial conditions and atmospheric forcings in this work.

To our knowledge, no study has been published yet, where the uncertainty of CLM carbon flux predictions has been comprehensively studied and estimated.

[revised manuscript text omitted]

perturbed atmospheric input data which resulted in a large ensemble spread of simulated NEE and LAI. One possible reason could be a too simplified and generalized phenology representation for those PFTs  (Dahlin et al. 2015). The stress deciduous phenology scheme (which determines plant onset and offset for C3 crop and C3 grass in CLM as outlined in Section 2.2) is strongly determined by various arbitrary thresholds such as "crit_offset_swi" (water stress days for offset trigger) or "crit_onset_fdd" (critical number of freezing degree days to trigger onset). Thus, probably model internal thresholds related to the meteorological input data such as temperature and soil moisture may result in changes in the overall carbon cycle.

[revised manuscript text omitted]

Peters, W., Jacobson, A.R., Sweeney, C., Andrews, A.E., Conway, T.J., Masarie, K., Miller, J.B., Bruhwiler, L.M.P., Pétron, G., Hirsch, A.I., Worthy, D.E.J., Werf, G.R. van der, Randerson, J.T., Wennberg, P.O., Krol, M.C., Tans, P.P., 2007. An atmospheric perspective on North American carbon dioxide exchange: CarbonTracker. PNAS 104, 18925–18930. doi:10.1073/pnas.0708986104

Peylin, P., Law, R.M., Gurney, K.R., Chevallier, F., Jacobson, A.R., Maki, T., Niwa, Y., Patra, P.K., Peters, W., Rayner, P.J., others, 2013. Global atmospheric carbon budget: results from an ensemble of atmospheric CO 2 inversions. Biogeosciences Discussions 10, 5301–5360.

[revised manuscript text omitted]

Tolk, L.F., Dolman, A.J., Meesters, A.G.C.A., Peters, W., 2011. A comparison of different inverse carbon flux estimation approaches for application on a regional domain. Atmos. Chem. Phys. 11, 10349–10365. doi:10.5194/acp-11-10349-2011

Tillack, A., Clasen, A., Kleinschmit, B., Förster, M., 2014. Estimation of the seasonal leaf area index in an alluvial forest using high-resolution satellite-based vegetation indices. Remote Sens. Environ. 141, 52–63.

Turner, D.P., Göckede, M., Law, B.E., Ritts, W.D., Cohen, W.B., Yang, Z., Hudiburg, T., Kennedy, R., Duane, M., 2011. Multiple constraint analysis of regional land surface carbon flux. Tellus B 63, 207–221. doi:10.1111/j.1600-0889.2011.00525.x

[revised manuscript text omitted]

---

## Author Response (AR2)

**Dear Sönke Zaehle,**

thank you very much for your final comments and suggestions and the quick response. We addressed all points as summarized below (in grey: editor comments, in black: our reply/ suggested changes).

**Abstract:**

- not clear what the NEE sum is. Do you mean to say annual NEE?

**Yes, we now replaced "NEE sum" by "annual NEE".**

- sentence beginning with "Thus, the parameter estimates would". This sentence seems redundant to me with the previous sentence. In any case, I don't understand the use of would here, either the new parameters change the carbon balance, as was said before, or they do not, but given your results, there should be a clear answer.

We did not estimate the carbon balance (including fossil fuel combustion, carbon fluxes due to land use changes, etc.) but just NEE, therefore "would". We reformulated the sentence now: "Thus, the estimation of CLM parameters with local NEE observations can be of high relevance when determining regional carbon balances."

- sentence beginning with "The increase of sigma\_sum\_NEE" is hard to understand. Either revise to be clear ("substantially lower than for crops") or remove.

**We replaced this sentence (Abstract, p.1, line 26ff) by:**

"The NEE uncertainty for the forest PFT was considerably lower ( $\sigma \ge$  NEE ~ 4.0 -13.5 gC m-2 y-1 with perturbed parameters, meteorological forcings and initial states). We conclude that LAI and NEE uncertainty with CLM is clearly underestimated if uncertain meteorological forcings and initial states are not taken into account."

**introduction**

P2 L2 NEE is the difference between soil AND PLANT respiration and photosynthetic CO2 uptake

**Correct, the "and plant" somehow got lost on the way, we added it again.**

P2 L17: use affect instead of determine (there are other factors as well)

**Done.**

P2 L21: Is Todd-Brown et al. 2013, which evaluates soil carbon stocks in ESMs really a good reference to suggest that the phenology of LSMs is flawed?

No, something was confused here. We now replaced the reference by "e.g. Richardson et al. 2012" and added "including CLM (Dahlin et al. 2015)".

P4 L23: remove duplicate ','b

Second "," is removed now.

**Additional changes:**

In chapter 2.3.1. (p.8,I.20) we added:

"DREAM(zs) is based on the Bayes theorem (A1)."

To stress the high impact of Q10 on the uncertainty of carbon fluxes for forest PFTs we added:

In the results (P.16, I.21-23 ff.):

"For the forest PFTs, the effect of  $Q_{10}$  (EnsP+Q10) on the uncertainty of the regional scale  $\Sigma$ GPPPFT,  $\Sigma$ ERPFT and  $\Sigma$ NEEPFT was much stronger than the effect of both uncertain meteorological forcings and initial states (CLM-EnsPAI)."

**In the discussion (P. 17 I.24 ff.):**

"The crucial role of the  $Q_{10}$  parameter for carbon stock and flux predictions and the respective uncertainties in most land surface models was already highlighted in previous studies (Post et al., 2008; Hararuk et al., 2014, Post et al., 2016). Post et al. (2016) showed that  $Q_{10}$  correlates strongly with other CLM4.5 key parameters like flNR and the Ball-Berry slope of stomatal conductance as well as with the initial carbon-nitrogen pools. This may explain why the spread of the regional scale  $\Sigma$ GPPPFT,  $\Sigma$ ERPFT and  $\Sigma$ NEEPFT in EnsP+Q10 was notably higher for coniferous forest than for the other PFTs (Figure 6). "

Changes in the wording of the last part in the conclusions (p.19, I 23ff):

"The effect of uncertainty of atmospheric forcing data on the LAI and NEE uncertainty was considerably higher for C3-grass and C3-crop than for the forest PFTs. This is related to the C3crop/C3-grass specific phenology representation and model internal thresholds in CLM. Many of these thresholds relate to temperature and strongly control leaf onset and senescence. The strong effect of the perturbed forcing data on modelled carbon fluxes in CLM4.5 is closely linked to the effect of uncertain model parameters like the temperature coefficient  $Q_{10}$ .

The model uncertainty resulting from uncertain atmospheric forcing data for C3-grass and C3-crop is additionally enhanced because these crops are located in the northern part of the catchment which is prone to some drought stress in summer. In combination with soil moisture related model thresholds, the effect of the perturbed precipitation on modelled NEE and LAI can be regionally different due to differences in the regional climate. This stresses the importance of regional scale modelling."